# Towards closing the gap between the theory and practice of SVRG

Othmane Sebbouh
LTCI, Télécom Paris
Institut Polytechnique de Paris
othmane.sebbouh@gmail.com

Nidham Gazagnadou
LTCI, Télécom Paris
Institut Polytechnique de Paris
nidham.gazagnadou@telecom-paris.fr

Samy Jelassi
ORFE Department
Princeton University
sjelassi@princeton.edu

Francis Bach
INRIA - Ecole Normale Supérieure
PSL Research University
francis.bach@inria.fr

Robert M. Gower
LTCI, Télécom Paris
Institut Polytechnique de Paris
robert.gower@telecom-paris.fr

## Abstract

Amongst the very first variance reduced stochastic methods for solving the empirical risk minimization problem was the SVRG method [13]. SVRG is an inner-outer loop based method, where in the outer loop a reference full gradient is evaluated, after which $m \in \mathbb{N}$ steps of an inner loop are executed where the reference gradient is used to build a variance reduced estimate of the current gradient. The simplicity of the SVRG method and its analysis have lead to multiple extensions and variants for even non-convex optimization. Yet there is a significant gap between the parameter settings that the analysis suggests and what is known to work well in practice. Our first contribution is that we take several steps towards closing this gap. In particular, the current analysis shows that $m$ should be of the order of the condition number so that the resulting method has a favorable complexity. Yet in practice $m = n$ works well regardless of the condition number, where $n$ is the number of data points. Furthermore, the current analysis shows that the inner iterates have to be reset using averaging after every outer loop. Yet in practice SVRG works best when the inner iterates are updated continuously and not reset. We provide an analysis of these aforementioned practical settings and show that they achieve the same favorable complexity as the original analysis (with slightly better constants). Our second contribution is to provide a more general analysis than had been previously done by using arbitrary sampling, which allows us to analyse virtually all forms of mini-batching through a single theorem. Since our setup and analysis reflect what is done in practice, we are able to set the parameters such as the mini-batch size and step size using our theory in such a way that produces a more efficient algorithm in practice, as we show in extensive numerical experiments.

# 1 Introduction

Consider the problem of minimizing a $\mu$–strongly convex and $L$–smooth function $f$ where

$$x^* = \arg\min_{x \in \mathbb{R}^d} \frac{1}{n} \sum_{i=1}^{n} f_i(x) =: f(x), \tag{1}$$

and each $f_i$ is convex and $L_i$–smooth. Several training problems in machine learning fit this format, e.g. least-squares, logistic regressions and conditional random fields. Typically each $f_i$ represents a regularized loss of an $i$th data point. When $n$ is large, algorithms that rely on full passes over the data, such as gradient descent, are no longer competitive. Instead, the stochastic version of gradient descent SGD [26] is often used since it requires only a mini-batch of data to make progress towards the solution. However, SGD suffers from high variance, which keeps the algorithm from converging unless a carefully often hand-tuned decreasing sequence of step sizes is chosen. This often results in a cumbersome parameter tuning and a slow convergence.

To address this issue, many variance reduced methods have been designed in recent years including SAG [27], SAGA [6] and SDCA [28] that require only a constant step size to achieve linear convergence. In this paper, we are interested in variance reduced methods with an inner-outer loop structure, such as S2GD [14], SARAH [21], L-SVRG [16] and the orignal SVRG [13] algorithm. Here we present not only a more general analysis that allows for any mini-batching strategy, but also a more *practical* analysis, by analysing methods that are based on what works in practice, and thus providing an analysis that can inform practice.

# 2 Background and Contributions

**Convergence under arbitrary samplings.**
We give the first arbitrary sampling convergence results for SVRG type methods in the convex setting[1]. That is our analysis includes all forms of sampling including mini-batching and importance sampling as a special case. To better understand the significance of this result, we use mini-batching $b$ elements *without* replacement as a running example throughout the paper. With this sampling the update step of SVRG, starting from $x^0 = w_0 \in \mathbb{R}^d$, takes the form of

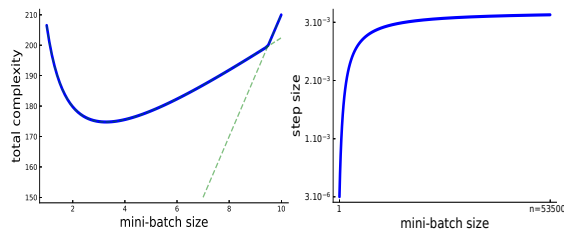

Figure 1: Left: the total complexity (3) for random Gaussian data, right: the step size (4) as $b$ increases.

$$x^{t+1} = x^t - \alpha \left( \frac{1}{b} \sum_{i \in B} \nabla f_i(x^t) - \frac{1}{b} \sum_{i \in B} \nabla f_i(w_{s-1}) + \nabla f(w_{s-1}) \right), \tag{2}$$

where $\alpha > 0$ is the step size, $B \subseteq [n] \stackrel{\text{def}}{=} \{1, \ldots, n\}$ and $b = |B|$. Here $w_{s-1}$ is the *reference point* which is updated after $m \in \mathbb{N}$ steps, the $x^t$'s are the *inner iterates* and $m$ is the loop length. As a special case of our forthcoming analysis in Corollary 4.1, we show that the *total complexity* of the SVRG method based on (2) to reach an $\epsilon > 0$ accurate solution has a simple expression which depends on $n$, $m$, $b$, $\mu$, $L$ and $L_{\max} \stackrel{\text{def}}{=} \max_{i \in [n]} L_i$:

$$C_m(b) \stackrel{\text{def}}{=} 2 \left( \frac{n}{m} + 2b \right) \max \left\{ \frac{3}{b} \frac{n-b}{n-1} \frac{L_{\max}}{\mu} + \frac{n}{b} \frac{b-1}{n-1} \frac{L}{\mu}, m \right\} \log \left( \frac{1}{\epsilon} \right), \tag{3}$$

so long as the step size is

$$\alpha = \frac{1}{2} \frac{b(n-1)}{3(n-b)L_{\max} + n(b-1)L}. \tag{4}$$

By total complexity we mean the total number of individual $\nabla f_i$ gradients evaluated. This shows that the total complexity is a simple function of the loop length $m$ and the mini-batch size $b$. See Figure 1 for an example for how total complexity evolves as we increase the mini-batch size.

**Optimal mini-batch and step sizes for SVRG.** The size of the mini-batch $b$ is often left as a parameter for the user to choose or set using a rule of thumb. The current analysis in the literature for mini-batching shows that when increasing the mini-batch size $b$, while the iteration complexity can decrease[2], the total complexity increases or is invariant. See for instance results in the non-convex case [22, 25], and for the convex case [10], [15], [1] and finally [18] where one can find the iteration complexity of several variants of SVRG with mini-batching. However, in practice, mini-batching can often lead to faster algorithms. In contrast our total complexity (3) clearly highlights that when increasing the mini batch size, the total complexity can decrease and the step size increases, as can be seen in our plot of (3) and (4) in Figure 1. Furthermore $C_m(b)$ is a convex function in $b$ which allows us to determine the optimal mini-batch a priori. For $m = n$ – a widely used loop length in practice – the optimal mini-batch size, depending on the problem setting, is given in Table 1. Moreover, we can also determine the optimal loop length. The reason we were able to achieve these

| $n \leq \frac{L}{\mu}$ | | $\frac{L}{\mu} < n < \frac{3L_{\max}}{\mu}$ | | |
|---|---|---|---|---|
| $L_{\max} \geq \frac{nL}{3}$ | $L_{\max} < \frac{nL}{3}$ | $L_{\max} \geq \frac{nL}{3}$ | $L_{\max} < \frac{nL}{3}$ | $n \geq \frac{3L_{\max}}{\mu}$ |
| $n$ | $\lceil \hat{b} \rceil$ | $\lceil \tilde{b} \rceil$ | $\lceil \min\{\hat{b}, \tilde{b}\} \rceil$ | $1$ |

Table 1: Optimal mini-batch sizes for Algorithm 1 with $m = n$. The last line presents the optimal mini-batch sizes depending on all the possible problem settings, which are presented in the first two lines. Notations: $\hat{b} = \sqrt{\frac{n}{2} \frac{3L_{\max} - L}{nL - 3L_{\max}}}$, $\tilde{b} = \frac{(3L_{\max} - L)n}{n(n-1)\mu - nL + 3L_{\max}}$.

new tighter mini-batch complexity bounds was by using the recently introduced concept of expected smoothness [9] alongside a new constant we introduce in this paper called the expected residual constant. The expected smoothness and residual constants, which we present later in Lemmas 4.1 and 4.2, show how mini-batching (and arbitrary sampling in general) combined with the smoothness of our data can determine how smooth in expectation our resulting mini-batched functions are. The expected smoothness constant has been instrumental in providing a tight mini-batch analysis for SGD [8], SAGA [7] and now SVRG.

**New practical variants.** We took special care so that our analysis allows for practical parameter settings. In particular, often the loop length is set to $m = n$ or $m = n/b$ in the case of mini-batching[3]. And yet, the classical SVRG analysis given in [13] requires $m \geq 20L_{\max}/\mu$ in order to ensure a resulting iteration complexity of $O((n + L_{\max}/\mu)\log(1/\epsilon))$. Furthermore, the standard SVRG analysis relies on averaging the $x^t$ inner iterates after every $m$ iterations of (2), yet this too is not what works well in practice[4]. To remedy this, we propose *Free-SVRG*, a variant of SVRG where the inner iterates are not averaged at any point. Furthermore, by developing a new Lyapunov style convergence for *Free-SVRG*, our analysis holds for any choice of $m$, and in particular, for $m = n$ we show that the resulting complexity is also given by $O((n + L_{\max}/\mu)\log(1/\epsilon))$.

The only downside of *Free-SVRG* is that the reference point is set using a weighted averaging based on the strong convexity parameter. To fix this issue, [11], and later [16, 17], proposed a loopless version of SVRG. This loopless variant has no explicit inner-loop structure, it instead updates the reference point based on a coin toss and lastly requires no knowledge of the strong convexity parameter and no averaging whatsoever. We introduce *L-SVRG-D*, an improved variant of *Loopless-SVRG* that takes much larger step sizes after the reference point is reset, and gradually smaller step sizes thereafter.

We provide an complexity analysis of *L-SVRG-D* that allows for arbitrary sampling and achieves the same complexity as *Free-SVRG*, albeit at the cost of introducing more variance into the procedure due to the coin toss.

## 3 Assumptions and Sampling

We collect all of the assumptions we use in the following.

**Assumption 3.1.** *There exist $L \geq 0$ and $\mu \geq 0$ such that for all $x, y \in \mathbb{R}^d$,*

$$
\begin{align}
f(x) &\leq f(y) + \langle \nabla f(y), x - y \rangle + \frac{L}{2} \|x - y\|_2^2, \tag{5} \\
f(x) &\leq f(y) + \langle \nabla f(x), x - y \rangle - \frac{\mu}{2} \|x - y\|_2^2. \tag{6}
\end{align}
$$

*We say that $f$ is $L$–smooth (5) and $\mu$–strongly convex (6). Moreover, for all $i \in [n]$, $f_i$ is convex and there exists $L_i \geq 0$ such that $f_i$ is $L_i$–smooth.*

So that we can analyse all forms of mini-batching simultaneously through arbitrary sampling we make use of a *sampling vector*.

**Definition 3.1** (The sampling vector). *We say that the random vector $v = [v_1, \ldots, v_n] \in \mathbb{R}^n$ with distribution $\mathcal{D}$ is a sampling vector if $\mathbb{E}_{\mathcal{D}}[v_i] = 1$ for all $i \in [n]$.*

With a sampling vector we can compute an unbiased estimate of $f(x)$ and $\nabla f(x)$ via

$$
f_v(x) \stackrel{\text{def}}{=} \frac{1}{n} \sum_{i=1}^n v_i f_i(x) \quad \text{and} \quad \nabla f_v(w) \stackrel{\text{def}}{=} \frac{1}{n} \sum_{i=1}^n v_i \nabla f_i(x). \tag{7}
$$

Indeed these are unbiased estimators since

$$
\mathbb{E}_{\mathcal{D}}[f_v(x)] = \frac{1}{n} \sum_{i=1}^n \mathbb{E}_{\mathcal{D}}[v_i] f_i(x) = \frac{1}{n} \sum_{i=1}^n f_i(x) = f(x). \tag{8}
$$

Likewise we can show that $\mathbb{E}_{\mathcal{D}}[\nabla f_v(x)] = \nabla f(x)$. Computing $\nabla f_v$ is cheaper than computing the full gradient $\nabla f$ whenever $v$ is a sparse vector. In particular, this is the case when the support of $v$ is based on a mini-batch sampling.

**Definition 3.2** (Sampling). *A sampling $S \subseteq [n]$ is any random set-valued map which is uniquely defined by the probabilities $\sum_{B \subseteq [n]} p_B = 1$ where $p_B \stackrel{\text{def}}{=} \mathbb{P}(S = B)$ for all $B \subseteq [n]$. A sampling $S$ is called proper if for every $i \in [n]$, we have that $p_i \stackrel{\text{def}}{=} \mathbb{P}[i \in S] = \sum_{C: i \in C} p_C > 0$.*

We can build a sampling vector using sampling as follows.

**Lemma 3.1** (Sampling vector). *Let $S$ be a proper sampling. Let $p_i \stackrel{\text{def}}{=} \mathbb{P}[i \in S]$ and $\mathbf{P} \stackrel{\text{def}}{=} \text{Diag}(p_1, \ldots, p_n)$. Let $v = v(S)$ be a random vector defined by*

$$
v(S) = \mathbf{P}^{-1} \sum_{i \in S} e_i \stackrel{\text{def}}{=} \mathbf{P}^{-1} e_S. \tag{9}
$$

*It follows that $v$ is a sampling vector.*

*Proof.* The $i$-th coordinate of $v(S)$ is $v_i(S) = \mathbb{1}(i \in S)/p_i$ and thus

$$
\mathbb{E}[v_i(S)] = \frac{\mathbb{E}[\mathbb{1}(i \in S)]}{p_i} = \frac{\mathbb{P}[i \in S]}{p_i} = 1. \qquad \square
$$

Our forthcoming analysis holds for all samplings. However, we will pay particular attention to $b$-nice sampling, otherwise known as mini-batching without replacement, since it is often used in practice.

**Definition 3.3** ($b$-nice sampling). *$S$ is a $b$-nice sampling if it is sampling such that*

$$
\mathbb{P}[S = B] = \frac{1}{\binom{n}{b}}, \quad \forall B \subseteq [n] : |B| = b.
$$

To construct such a sampling vector based on the $b$–nice sampling, note that $p_i = \frac{b}{n}$ for all $i \in [n]$ and thus we have that $v(S) = \frac{n}{b} \sum_{i \in S} e_i$ according to Lemma 3.1. The resulting subsampled function is then $f_v(x) = \frac{1}{|S|} \sum_{i \in S} f_i(x)$, which is simply the mini-batch average over $S$.

Using arbitrary sampling also allows us to consider non-uniform samplings, and for completeness, we present this sampling and several others in Appendix D.

## 4  *Free-SVRG*: freeing up the inner loop size

Similarly to SVRG, *Free-SVRG* is an inner-outer loop variance reduced algorithm. It differs from the original SVRG [13] on two major points: how the reference point is reset and how the first iterate of the inner loop is defined, see Algorithm 1[5].

First, in SVRG, the reference point is the average of the iterates of the inner loop. Thus, old iterates and recent iterates have equal weights in the average. This is counterintuitive as one would expect that to reduce the variance of the gradient estimate used in (2), one needs a reference point which is closer to the more recent iterates. This is why, inspired by [20], we use the weighted averaging in *Free-SVRG* given in (10), which gives more importance to recent iterates compared to old ones.

Second, in SVRG, the first iterate of the inner loop is reset to the reference point. Thus, the inner iterates of the algorithm are not updated using a one step recurrence. In contrast, *Free-SVRG* defines the first iterate of the inner loop as the last iterate of the previous inner loop, as is also done in practice. These changes and a new Lyapunov function analysis are what allows us to freely choose the size of the inner loop[6]. To declutter the notation, we define for a given step size $\alpha > 0$:

$$S_m \overset{\text{def}}{=} \sum_{i=0}^{m-1} (1 - \alpha\mu)^{m-1-i} \quad \text{and} \quad p_t \overset{\text{def}}{=} \frac{(1 - \alpha\mu)^{m-1-t}}{S_m}, \quad \text{for } t = 0, \ldots, m-1. \tag{10}$$

---

**Algorithm 1** *Free-SVRG*

---

**Parameters** inner-loop length $m$, step size $\alpha$, a sampling vector $v \sim \mathcal{D}$, and $p_t$ defined in (10)
**Initialization** $w_0 = x_0^m \in \mathbb{R}^d$
**for** $s = 1, 2, \ldots, S$ **do**

   $x_s^0 = x_{s-1}^m$
   **for** $t = 0, 1, \ldots, m-1$ **do**

      Sample $v_t \sim \mathcal{D}$
      $g_s^t = \nabla f_{v_t}(x_s^t) - \nabla f_{v_t}(w_{s-1}) + \nabla f(w_{s-1})$
      $x_s^{t+1} = x_s^t - \alpha g_s^t$
   $w_s = \sum_{t=0}^{m-1} p_t x_s^t$
**return** $x_S^m$

---

### 4.1  Convergence analysis

Our analysis relies on two important constants called the *expected smoothness* constant and the *expected residual* constant. Their existence is a result of the smoothness of the function $f$ and that of the individual functions $f_i, i \in [n]$.

**Lemma 4.1** (Expected smoothness, Theorem 3.6 in [8]). *Let $v \sim \mathcal{D}$ be a sampling vector and assume that Assumption 3.1 holds. There exists $\mathcal{L} \geq 0$ such that for all $x \in \mathbb{R}^d$,*

$$\mathbb{E}_{\mathcal{D}} \left[ \|\nabla f_v(x) - \nabla f_v(x^*)\|_2^2 \right] \leq 2\mathcal{L} \left( f(x) - f(x^*) \right). \tag{11}$$

**Lemma 4.2** (Expected residual). *Let $v \sim \mathcal{D}$ be a sampling vector and assume that Assumption 3.1 holds. There exists $\rho \geq 0$ such that for all $x \in \mathbb{R}^d$,*

$$\mathbb{E}_{\mathcal{D}} \left[ \|\nabla f_v(x) - \nabla f_v(x^*) - \nabla f(x)\|_2^2 \right] \leq 2\rho \left( f(x) - f(x^*) \right). \tag{12}$$

For completeness, the proof of Lemma 4.1 is given in Lemma E.1 in the supplementary material. The proof of Lemma 4.2 is also given in the supplementary material, in Lemma F.1. Indeed, all proofs are deferred to the supplementary material.

Though Lemma 4.1 establishes the existence of the expected smoothness $\mathcal{L}$, it was only very recently that a tight estimate of $\mathcal{L}$ was conjectured in [7] and proven in [8]. In particular, for our working example of $b$–nice sampling, we have that the constants $\mathcal{L}$ and $\rho$ have simple closed formulae that depend on $b$.

**Lemma 4.3** ($\mathcal{L}$ and $\rho$ for $b$-nice sampling). *Let $v$ be a sampling vector based on the $b$–nice sampling. It follows that.*

$$\mathcal{L} = \mathcal{L}(b) \overset{def}{=} \frac{1}{b}\frac{n-b}{n-1}L_{\max} + \frac{n}{b}\frac{b-1}{n-1}L, \tag{13}$$

$$\rho = \rho(b) \overset{def}{=} \frac{1}{b}\frac{n-b}{n-1}L_{\max}. \tag{14}$$

The reason that the expected smoothness and expected residual constants are so useful in obtaining a tight mini-batch analysis is because, as the mini-batch size $b$ goes from $n$ to $1$, $\mathcal{L}(b)$ (resp. $\rho(b)$) gracefully interpolates between the smoothness of the full function $\mathcal{L}(n) = L$ (resp. $\rho(n) = 0$), and the smoothness of the individual $f_i$ functions $\mathcal{L}(1) = L_{\max}$ (resp $\rho(1) = L_{\max}$). Also, we can bound the second moment of a variance reduced gradient estimate using $\mathcal{L}$ and $\rho$ as follows.

**Lemma 4.4.** *Let Assumption 3.1 hold. Let $x, w \in \mathbb{R}^d$ and $v \sim \mathcal{D}$ be sampling vector. Consider $g(x,w) \overset{def}{=} \nabla f_v(x) - \nabla f_v(w) + \nabla f(w)$. As a consequence of (11) and (12) we have that*

$$\mathbb{E}_{\mathcal{D}}\left[\|g(x,w)\|_2^2\right] \leq 4\mathcal{L}(f(x) - f(x^*)) + 4\rho(f(w) - f(x^*)). \tag{15}$$

Next we present a new Lyapunov style convergence analysis through which we will establish the convergence of the iterates and the function values simultaneously.

**Theorem 4.1.** *Consider the setting of Algorithm 1 and the following Lyapunov function*

$$\phi_s \overset{def}{=} \|x_s^m - x^*\|_2^2 + \psi_s \quad where \quad \psi_s \overset{def}{=} 8\alpha^2\rho S_m(f(w_s) - f(x^*)). \tag{16}$$

*If Assumption 3.1 holds and if $\alpha \leq \frac{1}{2(\mathcal{L}+2\rho)}$, then*

$$\mathbb{E}\left[\phi_s\right] \leq \beta^s\phi_0, \quad where \quad \beta = \max\left\{(1-\alpha\mu)^m, \frac{1}{2}\right\}. \tag{17}$$

## 4.2 Total complexity for $b$–nice sampling

To gain better insight into the convergence rate stated in Theorem 4.1, we present the total complexity of Algorithm 1 when $v$ is defined via the $b$–nice sampling introduced in Definition 3.3.

**Corollary 4.1.** *Consider the setting of Algorithm 1 and suppose that we use $b$–nice sampling. Let $\alpha = \frac{1}{2(\mathcal{L}(b)+2\rho(b))}$, where $\mathcal{L}(b)$ and $\rho(b)$ are given in (13) and (14). We have that the total complexity of finding an $\epsilon > 0$ approximate solution that satisfies $\mathbb{E}\left[\|x_s^m - x^*\|_2^2\right] \leq \epsilon\phi_0$ is*

$$C_m(b) \overset{def}{=} 2\left(\frac{n}{m} + 2b\right)\max\left\{\frac{\mathcal{L}(b) + 2\rho(b)}{\mu}, m\right\}\log\left(\frac{1}{\epsilon}\right). \tag{18}$$

Now (3) results from plugging (13) and (14) into (18). As an immediate sanity check, we check the two extremes $b = n$ and $b = 1$. When $b = n$, we would expect to recover the iteration complexity of gradient descent, as we do in the next corollary[7].

**Corollary 4.2.** *Consider the setting of Corollary 4.1 with $b = n$ and $m = 1$, thus $\alpha = \frac{1}{2(\mathcal{L}(n)+2\rho(n))} = \frac{1}{2L}$. Hence, the resulting total complexity (18) is given by $C_1(n) = 6n\frac{L}{\mu}\log\left(\frac{1}{\epsilon}\right)$.*

In practice, the most common setting is choosing $b = 1$ and the size of the inner loop $m = n$. Here we recover a complexity that is common to other non-accelerated algorithms [27], [6], [14], and for a range of values of $m$ including $m = n$.

**Corollary 4.3.** *Consider the setting of Corollary 4.1 with $b = 1$ and thus $\alpha = \frac{1}{2(\mathcal{L}(1)+2\rho(1))} = \frac{1}{6L_{\max}}$. Hence the resulting total complexity* (18) *is given by $C_m(1) = 18\left(n + \frac{L_{\max}}{\mu}\right)\log\left(\frac{1}{\epsilon}\right)$, so long as $m \in \left[\min(n, \frac{L_{\max}}{\mu}), \max(n, \frac{L_{\max}}{\mu})\right]$.*

Thus total complexity is essentially invariant for $m = n$, $m = L_{\max}/\mu$ and everything in between.

## 5  *L-SVRG-D*: a decreasing step size approach

Although *Free-SVRG* solves multiple issues regarding the construction and analysis of SVRG, it still suffers from an important issue: it requires the knowledge of the strong convexity constant, as is the case for the original SVRG algorithm [13]. One can of course use an explicit small regularization parameter as a proxy, but this can result in a slower algorithm.

A loopless variant of SVRG was proposed and analysed in [11, 16, 17]. At each iteration, their method makes a coin toss. With (a low) probability $p$, typically $1/n$, the reference point is reset to the previous iterate, and with probability $1 - p$, the reference point remains the same. This method does not require knowledge of the strong convexity constant.

Our method, *L-SVRG-D*, uses the same loopless structure as in [11, 16, 17] but introduces different step sizes at each iteration, see Algorithm 2. We initialize the step size to a fixed value $\alpha > 0$. At each iteration we toss a coin, and if it lands heads (with probability $1 - p$) the step size decreases by a factor $\sqrt{1-p}$. If it lands tails (with probability $p$) the reference point is reset to the most recent iterate and the step size is reset to its initial value $\alpha$.

This allows us to take larger steps than *L-SVRG* when we update the reference point, *i.e.,* when the variance of the unbiased estimate of the gradient is low, and smaller steps when this variance increases.

---

**Algorithm 2** L-SVRG-D

**Parameters** step size $\alpha$, $p \in (0, 1]$, and a sampling vector $v \sim \mathcal{D}$
**Initialization** $w^0 = x^0 \in \mathbb{R}^d$, $\alpha_0 = \alpha$
**for** $k = 1, 2, \ldots, K - 1$ **do**

　　Sample $v_k \sim \mathcal{D}$
　　$g^k = \nabla f_{v_k}(x^k) - \nabla f_{v_k}(w^k) + \nabla f(w^k)$
　　$x^{k+1} = x^k - \alpha_k g^k$
　　$(w^{k+1}, \alpha_{k+1}) = \begin{cases} (x^k, \alpha) & \text{with probability } p \\ (w^k, \sqrt{1-p}\,\alpha_k) & \text{with probability } 1 - p \end{cases}$

**return** $x^K$

---

**Theorem 5.1.** *Consider the iterates of Algorithm 2 and the following Lyapunov function*

$$\phi^k \stackrel{def}{=} \left\|x^k - x^*\right\|_2^2 + \psi^k \quad \text{where} \quad \psi^k \stackrel{def}{=} \frac{8\alpha_k^2 \mathcal{L}}{p(3 - 2p)}\left(f(w^k) - f(x^*)\right), \quad \forall k \in \mathbb{N}. \tag{19}$$

*If Assumption 3.1 holds and*

$$\alpha \leq \frac{1}{2\zeta_p \mathcal{L}}, \quad \text{where} \quad \zeta_p \stackrel{def}{=} \frac{(7 - 4p)(1 - (1-p)^{\frac{3}{2}})}{p(2 - p)(3 - 2p)}, \tag{20}$$

*then*

$$\mathbb{E}\left[\phi^k\right] \leq \beta^k \phi^0, \quad \text{where} \quad \beta = \max\left\{1 - \frac{2}{3}\alpha\mu, 1 - \frac{p}{2}\right\}. \tag{21}$$

**Remark 5.1.** *To get a sense of the formula of the step size given in* (20)*, it is easy to show that $\zeta_p$ is an increasing function of $p$ such that $7/4 \leq \zeta_p \leq 3$. Since typically $p \approx 0$, we often take a step which is approximately $\alpha \leq 2/(7\mathcal{L})$.*

**Corollary 5.1.** *Consider the setting of Algorithm 2 and suppose that we use b–nice sampling. Let* $\alpha = \frac{1}{2\zeta_p \mathcal{L}(b)}$. *We have that the total complexity of finding an* $\epsilon > 0$ *approximate solution that satisfies* $\mathbb{E}\left[\left\|x^k - x^*\right\|_2^2\right] \leq \epsilon\,\phi^0$ *is*

$$C_p(b) \stackrel{def}{=} 2(2b + pn)\max\left\{\frac{3\zeta_p}{2}\frac{\mathcal{L}(b)}{\mu}, \frac{1}{p}\right\}\log\left(\frac{1}{\epsilon}\right). \tag{22}$$

## 6 Optimal parameter settings: loop, mini-batch and step sizes

In this section, we restrict our analysis to $b$–nice sampling. First, we determine the optimal loop size for Algorithm 1. Then, we examine the optimal mini-batch and step sizes for particular choices of the inner loop size $m$ for Algorithm 1 and of the probability $p$ of updating the reference point in Algorithm 2, that play analogous roles. Note that the steps used in our algorithms depend on $b$ through the expected smoothness constant $\mathcal{L}(b)$ and the expected residual constant $\rho(b)$. Hence, optimizing the total complexity in the mini-batch size also determines the optimal step size.

Examining the total complexities of Algorithms 1 and 2, given in (18) and (22), we can see that, when setting $p = 1/m$ in Algorithm 2, these complexities only differ by constants. Thus, to avoid redundancy, we present the optimal mini-batch sizes for Algorithm 2 in Appendix C and we only consider here the complexity of Algorithm 1 given in (18).

### 6.1 Optimal loop size for Algorithm 1

Here we determine the optimal value of $m$ for a fixed batch size $b$, denoted by $m^*(b)$, which minimizes the total complexity (18).

**Proposition 6.1.** *The loop size that minimizes* (18) *and the resulting total complexity is given by*

$$m^*(b) = \frac{\mathcal{L}(b) + 2\rho(b)}{\mu} \quad and \quad C_{m^*}(b) = 2\left(n + 2b\frac{\mathcal{L}(b) + 2\rho(b)}{\mu}\right)\log\left(\frac{1}{\epsilon}\right). \tag{23}$$

For example when $b = 1$, we have that $m^*(1) = 3L_{\max}/\mu$ and $C_{m^*}(1) = O((n + L_{\max}/\mu)\log(1/\epsilon))$, which is the same complexity as achieved by the range of $m$ values given in Corollary 4.3. Thus, as we also observed in Corollary 4.3, the total complexity is not very sensitive to the choice of $m$, and $m = n$ is a perfectly safe choice as it achieves the same complexity as $m^*$. We also confirm this numerically with a series of experiments in Section G.2.2.

### 6.2 Optimal mini-batch and step sizes

In the following proposition, we determine the optimal mini-batch and step sizes for two practical choices of the size of the loop $m$.

**Proposition 6.2.** *Let* $b^* \stackrel{def}{=} \underset{b\in[n]}{\arg\min}\, C_m(b)$, *where* $C_m(b)$ *is defined in* (18). *For the widely used choice* $m = n$, *we have that* $b^*$ *is given by Table 1. For another widely used choice* $m = n/b$, *which allows to make a full pass over the data set during each inner loop, we have*

$$b^* = \begin{cases} \lfloor\bar{b}\rfloor & if\ n > \frac{3L_{\max}}{\mu} \\ 1 & if\ \frac{3L_{\max}}{L} < n \leq \frac{3L_{\max}}{\mu} \\ n & otherwise,\ if\ n \leq \frac{3L_{\max}}{L} \end{cases}, \quad where\ \bar{b} \stackrel{def}{=} \frac{n(n-1)\mu - 3n(L_{\max} - L)}{3(nL - L_{\max})}. \tag{24}$$

Previously, theory showed that the total complexity would increase as the mini-batch size increases, and thus established that single-element sampling was optimal. However, notice that for $m = n$ and $m = n/b$, the usual choices for $m$ in practice, the optimal mini-batch size is different than 1 for a range of problem settings. Since our algorithms are closer to the SVRG variants used in practice, we argue that our results explain why practitioners experiment that mini-batching works, as we verify in the next section.

# 7 Experiments

We performed a series of experiments on data sets from LIBSVM [5] and the UCI repository [3], to validate our theoretical findings. We tested $l_2$–regularized logistic regression on *ijcnn1* and *real-sim*, and ridge regression on *slice* and *YearPredictionMSD*. We used two choices for the regularizer: $\lambda = 10^{-1}$ and $\lambda = 10^{-3}$. All of our code is implemented in `Julia 1.0`. Due to lack of space, most figures have been relegated to Section G in the supplementary material.

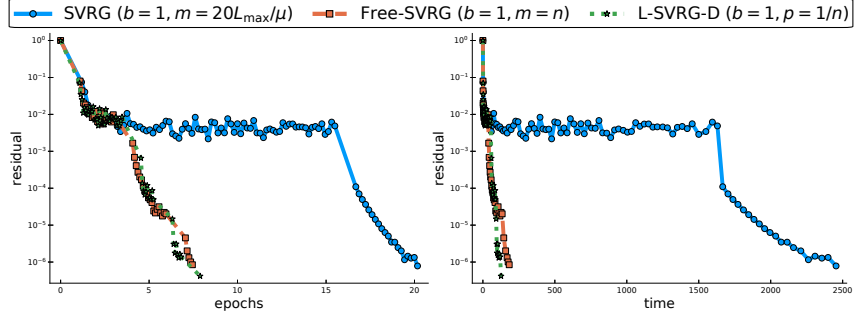

Figure 2: Comparison of theoretical variants of SVRG without mini-batching ($b = 1$) on the *ijcnn1* data set.

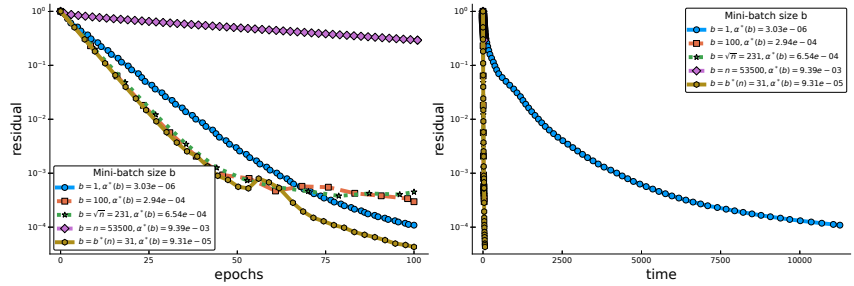

Figure 3: Optimality of our mini-batch size $b^*$ given in Table 1 for *Free-SVRG* on the *slice* data set.

**Practical theory.** Our first round of experiments aimed at verifying if our theory does result in efficient algorithms. Indeed, we found that *Free-SVRG* and *L-SVRG-D* with the parameter setting given by our theory are often faster than SVRG with settings suggested by the theory in [13], that is $m = 20L_{\max}/\mu$ and $\alpha = 1/10L_{\max}$. See Figure 2, and Section G.1 for more experiments comparing different theoretical parameter settings.

**Optimal mini-batch size.** We also confirmed numerically that when using *Free-SVRG* with $m = n$, the optimal mini-batch size $b^*$ derived in Table 1 was highly competitive as compared to the range of mini-batch sizes $b \in \{1, 100, \sqrt{n}, n\}$. See Figure 3 and several more such experiments in Section G.2.1. We also explore the optimality of our $m^*$ in more experiments in Section G.2.2.

**Acknowledgments**

RMG acknowledges the support by grants from DIM Math Innov Région Ile-de-France (ED574 - FMJH), reference ANR-11-LABX-0056-LMH, LabEx LMH.

## Footnotes

[1]SVRG has very recently been analysed under arbitrary samplings in the non-convex setting [12].

[2]Note that the total complexity is equal to the iteration complexity times the mini-batch size $b$.

[3]See for example the lightning package from `scikit-learn` [23]: http://contrib.scikit-learn.org/lightning/ and [21] for examples where $m = n$. See [2] for an example where $m = 5n/b$.

[4]Perhaps an exception to the above issues in the literature is the Katyusha method and its analysis [1], which is an accelerated variant of SVRG. In [1] the author shows that using a loop length $m = 2n$ and by not averaging the inner iterates, the Katyusha method achieves the accelerated complexity of $O((n + \sqrt{(nL_{\max})/\mu})\log(1/\epsilon))$. Though a remarkable advance in the theory of accelerated methods, the analysis in [1] does not hold for the unaccelerated case. This is important since, contrary to the name, the accelerated variants of stochastic methods are not always faster than their non-accelerated counterparts. Indeed, acceleration only helps in the stochastic setting when $L_{\max}/\mu \geq n$, in other words for problems that are sufficiently ill-conditioned.

[5]After submitting this work, it has come to our attention that *Free-SVRG* is a special case of *k-SVRG* [24] when $k = 1$.

[6]Hence the name of our method *Free*-SVRG.

[7]Though the resulting complexity is 6 times the tightest gradient descent complexity, it is of the same order.

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
