[Supplementary Material]

This is the supplementary material for the paper: "Towards closing the gap between the theory and practice of SVRG" authored by O. Sebbouh, N. Gazagnadou, S. Jelassi, F. Bach and R. M. Gower (NeurIPS 2019).

In Section A we present general properties that are used in our proofs. In Section B, we present the proofs for the convergence and the complexities of our algorithms. In Section D, we define several samplings. In Section E, we present the expected smoothness constant for the samplings we consider. In Section F, we present the expected residual constant for the same samplings.

## A   General properties

**Lemma A.1.** *For all $a, b \in \mathbb{R}^d$, $\|a + b\|_2^2 \leq 2\|a\|_2^2 + 2\|b\|_2^2$.*

**Lemma A.2.** *For any random vector $X \in \mathbb{R}^d$,*

$$\mathbb{E}\left[\|X - \mathbb{E}[X]\|_2^2\right] = \mathbb{E}\left[\|X\|_2^2\right] - \|\mathbb{E}[X]\|_2^2 \leq \mathbb{E}\left[\|X\|_2^2\right] \ .$$

**Lemma A.3.** *For any convex function $f$, we have*

$$f(y) \geq f(x) + \nabla f(x)^\top (y - x), \quad \forall x, y \in \mathbb{R}^d.$$

**Lemma A.4** (Logarithm inequality). *For all $x > 0$,*

$$\log(x) \leq x - 1 \ . \tag{25}$$

**Lemma A.5** (Complexity bounds). *Consider the sequence $(\alpha_k)_k \in \mathbb{R}_+$ of positive scalars that converges to $0$ according to*

$$\alpha_k \leq \rho^k \alpha_0,$$

*where $\rho \in [0, 1)$. For a given $\epsilon \in (0, 1)$, we have that*

$$k \geq \frac{1}{1 - \rho} \log\left(\frac{1}{\epsilon}\right) \implies \alpha_k \leq \epsilon\alpha_0. \tag{26}$$

**Lemma A.6.** *Consider convex and $L_i$–smooth functions $f_i$, where $L_i \geq 0$ for all $i \in [n]$, and define $L_{\max} = \max_{i \in [n]} L_i$. Let*

$$f(x) = \frac{1}{n} \sum_{i=1}^{n} f_i(x)$$

*for any $x \in \mathbb{R}^d$. Suppose that $f$ is $L$–smooth, where $L \geq 0$. Then,*

$$nL \geq L_{\max}. \tag{27}$$

*Proof.* Let $x, y \in \mathbb{R}^d$. Since $f$ is $L$–smooth, we have

$$f(x) \leq f(y) + \nabla f(y)^\top (x - y) + \frac{L}{2} \|x - y\|_2^2.$$

Hence, multiplying by $n$ on both sides,

$$\sum_{i=1}^{n} f_i(x) \leq \sum_{i=1}^{n} f_i(y) + \sum_{i=1}^{n} \nabla f_i(y)^\top (x - y) + \frac{nL}{2} \|x - y\|_2^2.$$

Rearranging this inequality,

$$\sum_{i=1}^{n} \left(f_i(x) - f_i(y) - \nabla f_i(y)^\top (x - y)\right) \leq \frac{nL}{2} \|x - y\|_2^2. \tag{28}$$

Since the functions $f_i$ are convex, we have for all $i \in [n]$,

$$f_i(x) - f_i(y) - \nabla f_i(y)^\top (x - y) \geq 0.$$

Then, as a consequence of (28), we have that for all $i \in [n]$,

$$f_i(x) - f_i(y) - \nabla f_i(y)^\top (x - y) \leq \frac{nL}{2} \|x - y\|_2^2.$$

Rearranging this inequality,

$$f_i(x) \leq f_i(y) + \nabla f_i(y)^\top (x - y) + \frac{nL}{2} \|x - y\|_2^2.$$

But since for all $i \in [n]$, $L_i$ is the smallest positive constant that verifies

$$f_i(x) \leq f_i(y) + \nabla f_i(y)^\top (x - y) + \frac{L_i}{2} \|x - y\|_2^2,$$

we have for all $i \in [n]$, $L_i \leq nL$. Hence $L_{\max} \leq nL$. $\qquad\square$

# B    Proofs of the results of the main paper

In this section, we will use the abbreviations $\mathbb{E}_t[X] \overset{\text{def}}{=} \mathbb{E}[X|x^t, \ldots, x^0]$ for any random variable $X \in \mathbb{R}^d$ and iterates $(x^t)_{t \geq 0}$.

## B.1    Proof of Lemma 4.4

*Proof.*

$$
\begin{aligned}
\mathbb{E}_{\mathcal{D}}\left[\|g(x, w)\|_2^2\right] \;=\; & \mathbb{E}_{\mathcal{D}}\left[\|\nabla f_v(x) - \nabla f_v(x^*) + \nabla f_v(x^*) - \nabla f_v(w) + \nabla f(w)\|_2^2\right] \\
\overset{\text{Lem. } A.1}{\leq}\; & 2\mathbb{E}_{\mathcal{D}}\left[\|\nabla f_v(x) - \nabla f_v(x^*)\|_2^2\right] \\
& + 2\mathbb{E}_{\mathcal{D}}\left[\|\nabla f_v(w) - \nabla f_v(x^*) - \nabla f(w)\|_2^2\right] \\
\overset{(11)+(12)}{\leq}\; & 4\mathcal{L}(f(x) - f(x^*)) + 4\rho(f(w) - f(x^*)).
\end{aligned}
$$

$\qquad\square$

## B.2    Proof of Theorem 4.1

*Proof.* To clarify the notations, we recall that $g_s^t \overset{\text{def}}{=} g(x_s^t, w_{s-1})$. Then, we get

$$
\begin{aligned}
\mathbb{E}_t\left[\|x_s^{t+1} - x^*\|_2^2\right] \;=\; & \mathbb{E}_t\left[\|x_s^t - x^* - \alpha g_s^t\|_2^2\right] \\
=\; & \|x_s^t - x^*\|_2^2 - 2\alpha \mathbb{E}_t\left[g_s^t\right]^\top (x_s^t - x^*) + \alpha^2 \mathbb{E}_t\left[\|g_s^t\|_2^2\right] \\
\overset{(8)+(15)}{\leq}\; & \|x_s^t - x^*\|_2^2 - 2\alpha \nabla f(x_s^t)^\top (x_s^t - x^*) \\
& + 2\alpha^2 \left[2\mathcal{L}(f(x_s^t) - f(x^*)) + 2\rho(f(w_{s-1}) - f(x^*))\right] \\
\overset{(6)}{\leq}\; & (1 - \alpha\mu) \|x_s^t - x^*\|_2^2 - 2\alpha(1 - 2\alpha\mathcal{L})\left(f(x_s^t) - f(x^*)\right) \\
& + 4\alpha^2 \rho(f(w_{s-1}) - f(x^*)). \qquad\qquad (29)
\end{aligned}
$$

Note that since $\alpha \leq \frac{1}{2(\mathcal{L}+2\rho)}$ and $\rho \geq 0$, we have that

$$\alpha \overset{\text{Lemma E.3}}{\leq} \frac{1}{2\mu},$$

and consequently $(1 - \alpha\mu) > 0$. Thus by iterating (29) over $t = 0, \ldots, m - 1$ and taking the expectation, since $x_s^0 = x_{s-1}^m$, we obtain

$$
\begin{aligned}
\mathbb{E}\left[\|x_s^m - x^*\|_2^2\right] &\leq (1 - \alpha\mu)^m \, \mathbb{E}\left[\|x_{s-1}^m - x^*\|_2^2\right] \\
&\quad - 2\alpha(1 - 2\alpha\mathcal{L}) \sum_{t=0}^{m-1} (1 - \alpha\mu)^{m-1-t} \, \mathbb{E}\left[f(x_s^t) - f(x^*)\right] \\
&\quad + 4\alpha^2\rho\mathbb{E}\left[f(w_{s-1}) - f(x^*)\right] \sum_{t=0}^{m-1} (1 - \alpha\mu)^{m-1-t} \\
&\overset{(10)}{=} (1 - \alpha\mu)^m \, \mathbb{E}\left[\|x_{s-1}^m - x^*\|_2^2\right] - 2\alpha(1 - 2\alpha\mathcal{L})S_m \sum_{t=0}^{m-1} p_t \mathbb{E}\left[f(x_s^t) - f(x^*)\right] \\
&\quad + 4\alpha^2\rho S_m \mathbb{E}\left[f(w_{s-1}) - f(x^*)\right] \\
&\overset{(16)}{=} (1 - \alpha\mu)^m \, \mathbb{E}\left[\|x_{s-1}^m - x^*\|_2^2\right] - 2\alpha(1 - 2\alpha\mathcal{L})S_m \sum_{t=0}^{m-1} p_t \mathbb{E}\left[f(x_s^t) - f(x^*)\right] \\
&\quad + \frac{1}{2}\mathbb{E}\left[\psi_{s-1}\right].
\end{aligned}
$$
(30)

Weights $p_t$ are defined in (10). We note that $(1 - \alpha\mu) > 0$ implies that $p_t > 0$ for all $t = 0, \ldots, m-1$, and by construction we get $\sum_{t=0}^{m-1} p_t = 1$. Since $f$ is convex, we have by Jensen's inequality that

$$
\begin{aligned}
f(w_s) - f(x^*) &= f\left(\sum_{t=0}^{m-1} p_t x_s^t\right) - f(x^*) \\
&\leq \sum_{t=0}^{m-1} p_t (f(x_s^t) - f(x^*)).
\end{aligned}
$$
(31)

Consequently,

$$
\mathbb{E}\left[\psi_s\right] \overset{(16)+(31)}{\leq} 8\alpha^2\rho S_m \sum_{t=0}^{m-1} p_t \mathbb{E}\left[(f(x_s^t) - f(x^*))\right].
$$
(32)

As a result,

$$
\begin{aligned}
\mathbb{E}\left[\phi_s\right] &= \mathbb{E}\left[\|x_s^m - x^*\|_2^2\right] + \mathbb{E}\left[\psi_s\right] \\
&\overset{(30)+(32)}{\leq} (1 - \alpha\mu)^m \, \mathbb{E}\left[\|x_{s-1}^m - x^*\|_2^2\right] + \frac{1}{2}\mathbb{E}\left[\psi_{s-1}\right] \\
&\quad - 2\alpha(1 - 2\alpha(\mathcal{L} + 2\rho))S_m \sum_{t=0}^{m-1} p_t \mathbb{E}\left[(f(x_s^t) - f(x^*))\right].
\end{aligned}
$$

Since $\alpha \leq \frac{1}{2(\mathcal{L}+2\rho)}$, the above implies

$$
\begin{aligned}
\mathbb{E}\left[\phi_s\right] &\leq (1 - \alpha\mu)^m \, \mathbb{E}\left[\|x_{s-1}^m - x^*\|_2^2\right] + \frac{1}{2}\mathbb{E}\left[\psi_{s-1}\right] \\
&\leq \beta\mathbb{E}\left[\phi_{s-1}\right],
\end{aligned}
$$

where $\beta = \max\{(1 - \alpha\mu)^m, \frac{1}{2}\}$.

Moreover, if we set $w_s = x_s^t$ with probability $p_t$, for $t = 0, \ldots, m-1$, the result would still hold. Indeed (31) would hold with equality and the rest of the proof would follow verbatim. $\qquad\square$

## B.3  Proof of Corollary 4.1

*Proof.* Noting $\beta = \max\left\{\left(1 - \frac{\mu}{2(\mathcal{L}(b)+2\rho(b))}\right)^m, \frac{1}{2}\right\}$, we need to chose $s$ so that $\beta^s \leq \epsilon$, that is $s \geq \frac{\log(1/\epsilon)}{\log(1/\beta)}$. Since in each inner iteration we evaluate $2b$ gradients of the $f_i$ functions, and in each

outer iteration we evaluate all $n$ gradients, this means that the total complexity will be given by

$$C \overset{\text{def}}{=} (n + 2bm)\frac{\log(1/\epsilon)}{\log(1/\beta)}$$

$$= (n + 2bm)\max\left\{-\frac{1}{m\log(1 - \frac{\mu}{2(\mathcal{L}(b)+2\rho(b))})}, \frac{1}{\log 2}\right\}\log\left(\frac{1}{\epsilon}\right)$$

$$\overset{(25)}{\leq} (n + 2bm)\max\left\{\frac{1}{m}\frac{2(\mathcal{L}(b) + 2\rho(b))}{\mu}, 2\right\}\log\left(\frac{1}{\epsilon}\right).$$

$\square$

## B.4 Proof of Corollary 4.3

*Proof.* Recall that from (18), using the fact that $\mathcal{L}(1) = \rho(1) = L_{\max}$, we have

$$C_m(1) = 2\left(\frac{n}{m} + 2\right)\max\left\{\frac{3L_{\max}}{\mu}, m\right\}\log\left(\frac{1}{\epsilon}\right).$$

When $n \geq \frac{L_{\max}}{\mu}$, then, $m \in \left[\frac{L_{\max}}{\mu}, n\right]$. We can rewrite $C_m(1)$ as

$$C_m(1) = 2(n + 2m)\max\left\{\frac{1}{m}\frac{3L_{\max}}{\mu}, 1\right\}\log\left(\frac{1}{\epsilon}\right).$$

We have $\frac{1}{m}\frac{3L_{\max}}{\mu} \leq 3$ and $n + 2m \leq 3n$. Hence,

$$C_m(1) \leq 18n\log\left(\frac{1}{\epsilon}\right) = O\left(\left(n + \frac{L_{\max}}{\mu}\right)\log\left(\frac{1}{\epsilon}\right)\right).$$

When $n \leq \frac{L_{\max}}{\mu}$, then, $m \in \left[n, \frac{L_{\max}}{\mu}\right]$. We have $\frac{n}{m} \leq 1$ and $m \leq \frac{3L_{\max}}{\mu}$. Hence,

$$C_m(1) \leq \frac{18L_{\max}}{\mu}\log\left(\frac{1}{\epsilon}\right) = O\left(\left(n + \frac{L_{\max}}{\mu}\right)\log\left(\frac{1}{\epsilon}\right)\right).$$

$\square$

## B.5 Proof of Theorem 5.1

Before analysing Algorithm 2, we present a lemma that allows to compute the expectations $\mathbb{E}\left[\alpha_k\right]$ and $\mathbb{E}\left[\alpha_k^2\right]$, that will be used in the analysis.

**Lemma B.1.** *Consider the step sizes defined by Algorithm 2. We have*

$$\mathbb{E}\left[\alpha_k\right] = \frac{(1-p)^{\frac{3k+2}{2}}(1 - \sqrt{1-p}) + p}{1 - (1-p)^{\frac{3}{2}}}\alpha. \tag{33}$$

$$\mathbb{E}\left[\alpha_k^2\right] = \frac{1 + (1-p)^{2k+1}}{2 - p}\alpha^2. \tag{34}$$

*Proof.* Taking expectation with respect to the filtration induced by the sequence of step sizes $\{\alpha_1, \ldots, \alpha_k\}$

$$\mathbb{E}_p\left[\alpha_{k+1}\right] = (1-p)\sqrt{1-p}\,\alpha_k + p\alpha. \tag{35}$$

Then taking total expectation

$$\mathbb{E}\left[\alpha_{k+1}\right] = (1-p)\sqrt{1-p}\mathbb{E}\left[\alpha_k\right] + p\alpha. \tag{36}$$

Hence the sequence $(\mathbb{E}\left[\alpha_k\right])_{k\geq1}$ is uniquely defined by

$$\mathbb{E}\left[\alpha_k\right] = \frac{(1-p)^{\frac{3k+2}{2}}(1 - \sqrt{1-p}) + p}{1 - (1-p)^{\frac{3}{2}}}\alpha. \tag{37}$$

Indeed, applying (36) recursively gives

$$\mathbb{E}\left[\alpha_k\right] = (1-p)^{\frac{3k}{2}}\alpha + p\alpha\sum_{i=0}^{k-1}(1-p)^{\frac{3i}{2}}.$$

Adding up the geometric series gives

$$\begin{aligned}
\mathbb{E}\left[\alpha_k\right] &= \alpha(1-p)^{\frac{3k}{2}} + p\alpha\frac{1-(1-p)^{\frac{3k}{2}}}{1-(1-p)^{\frac{3}{2}}}\\
&= \frac{(1-p)^{\frac{3k}{2}}\left(1-(1-p)^{\frac{3}{2}}\right)-(1-p)^{\frac{3k}{2}}p+p}{1-(1-p)^{\frac{3}{2}}}\alpha .
\end{aligned}$$

Which leads to (37) by factorizing. The same arguments are used to compute $\mathbb{E}\left[\alpha_k^2\right]$. □

We now present a proof of Theorem 5.1.

*Proof.* We recall that $g^k \stackrel{\text{def}}{=} \nabla f(x^k)$. First, we get

$$\begin{aligned}
\mathbb{E}_k\left[\|x^{k+1}-x^*\|_2^2\right] &= \mathbb{E}_k\left[\|x^k-x^*-\alpha_k g^k\|_2^2\right]\\
&= \|x^k-x^*\|_2^2 - 2\alpha_k\mathbb{E}_k\left[g^k\right]^\top(x^k-x^*)+\alpha_k^2\mathbb{E}_k\left[\|g^k\|_2^2\right]\\
&\overset{(8)+(15)+\text{Rem. }E.1}{\leq} \|x^k-x^*\|_2^2 - 2\alpha_k\nabla f(x^k)^\top(x^k-x^*)\\
&\qquad +2\alpha_k^2\left[2\mathcal{L}(f(x^k)-f(x^*))+2\mathcal{L}(f(w^k)-f(x^*))\right]\\
&\overset{(6)}{\leq} (1-\alpha_k\mu)\|x^k-x^*\|_2^2 - 2\alpha_k(1-2\alpha_k\mathcal{L})\left(f(x^k)-f(x^*)\right)\\
&\qquad +4\alpha_k^2\mathcal{L}(f(w^k)-f(x^*))\\
&\overset{(19)}{=} (1-\alpha_k\mu)\|x^k-x^*\|_2^2 - 2\alpha_k(1-2\alpha_k\mathcal{L})\left(f(x^k)-f(x^*)\right)\\
&\qquad +p\left(\frac{3}{2}-p\right)\psi^k.
\end{aligned}$$

Hence we have, taking total expectation and noticing that the variables $\alpha_k$ and $x^k$ are independent,

$$\begin{aligned}
\mathbb{E}\left[\|x^{k+1}-x^*\|_2^2\right] &\leq (1-\mathbb{E}\left[\alpha_k\right]\mu)\mathbb{E}\left[\|x^k-x^*\|_2^2\right] - 2\mathbb{E}\left[\alpha_k(1-2\alpha_k\mathcal{L})\right]\mathbb{E}\left[f(x^k)-f(x^*)\right]\\
&\qquad +p\left(\frac{3}{2}-p\right)\mathbb{E}\left[\psi^k\right]. \tag{38}
\end{aligned}$$

We have also have

$$\begin{aligned}
\mathbb{E}_k\left[\psi^{k+1}\right] &= (1-p)\frac{8(1-p)\alpha_k^2\mathcal{L}}{p(3-2p)}\left(f(w^k)-f(x^*)\right)+p\frac{8\alpha^2\mathcal{L}}{p(3-2p)}\left(f(x^k)-f(x^*)\right)\\
&= (1-p)^2\psi^k + \frac{8\alpha^2\mathcal{L}}{3-2p}\left(f(x^k)-f(x^*)\right).
\end{aligned}$$

Hence, taking total expectation gives

$$\mathbb{E}\left[\psi^{k+1}\right] = (1-p)^2\mathbb{E}\left[\psi^k\right] + \frac{8\alpha^2\mathcal{L}}{3-2p}\mathbb{E}\left[f(x^k)-f(x^*)\right] \tag{39}$$

Consequently,

$$\mathbb{E}\left[\phi^{k+1}\right] \overset{(19)+(38)+(39)}{\leq} (1 - \mathbb{E}\left[\alpha_k\right]\mu)\,\mathbb{E}\left[\left\|x^k - x^*\right\|_2^2\right]$$
$$-2\left(\mathbb{E}\left[\alpha_k(1 - 2\alpha_k\mathcal{L})\right] - 4\frac{\alpha^2\mathcal{L}}{3 - 2p}\right)\mathbb{E}\left[f(x^k) - f(x^*)\right]$$
$$+\left(1 - \frac{p}{2}\right)\mathbb{E}\left[\psi^k\right]$$
$$= (1 - \mathbb{E}\left[\alpha_k\right]\mu)\,\mathbb{E}\left[\left\|x^k - x^*\right\|_2^2\right]$$
$$-2\left(\mathbb{E}\left[\alpha_k\right] - 2\left(\mathbb{E}\left[\alpha_k^2\right] + \frac{2}{3 - 2p}\alpha^2\right)\mathcal{L}\right)\mathbb{E}\left[f(x^k) - f(x^*)\right]$$
$$+\left(1 - \frac{p}{2}\right)\mathbb{E}\left[\psi^k\right]. \tag{40}$$

From Lemma B.1, we have $\mathbb{E}\left[\alpha_k\right] = \frac{(1-p)^{\frac{3k+2}{2}}(1-\sqrt{1-p})+p}{1-(1-p)^{\frac{3}{2}}}\alpha$, and we can show that for all $k$

$$\mathbb{E}\left[\alpha_k\right] \geq \frac{2}{3}\alpha, \tag{41}$$

Letting $q = 1 - p$ we have that

$$\frac{(1-p)^{\frac{3k+2}{2}}(1 - \sqrt{1-p}) + p}{1 - (1-p)^{\frac{3}{2}}} = \frac{q^{\frac{3k+2}{2}}(1 - \sqrt{q}) + 1 - q}{1 - q^{\frac{3}{2}}}$$
$$= q^{\frac{3k+2}{2}}\frac{1 - \sqrt{q}}{1 - q^{3/2}} + \frac{1 - q}{1 - q^{3/2}}$$
$$\geq \frac{1 - q}{1 - q^{3/2}}$$
$$\geq \frac{2}{3}, \quad \forall q \in [0, 1] .$$

Consequently,

$$\mathbb{E}\left[\phi^{k+1}\right] \overset{(38)+(39)+(41)}{\leq} \left(1 - \frac{2}{3}\alpha\mu\right)\mathbb{E}\left[\left\|x^k - x^*\right\|_2^2\right]$$
$$-2\left(\mathbb{E}\left[\alpha_k\right] - 2\left(\mathbb{E}\left[\alpha_k^2\right] + \frac{2}{3 - 2p}\alpha^2\right)\mathcal{L}\right)\mathbb{E}\left[f(x^k) - f(x^*)\right]$$
$$+\left(1 - \frac{p}{2}\right)\mathbb{E}\left[\psi^k\right]. \tag{42}$$

To declutter the notations, Let us define

$$a_k \overset{\text{def}}{=} \frac{(1-p)^{\frac{3k+2}{2}}(1 - \sqrt{1-p}) + p}{1 - (1-p)^{\frac{3}{2}}} \tag{43}$$

$$b_k \overset{\text{def}}{=} \frac{1 + (1-p)^{2k+1}}{2 - p} \tag{44}$$

so that $\mathbb{E}\left[\alpha_k\right] = a_k\alpha$ and $\mathbb{E}\left[\alpha_k^2\right] = b_k\alpha^2$. Then (42) becomes

$$\mathbb{E}\left[\phi^{k+1}\right] \leq \left(1 - \frac{2}{3}\alpha\mu\right)\mathbb{E}\left[\left\|x^k - x^*\right\|_2^2\right]$$
$$-2\alpha\left(a_k - 2\alpha\left(b_k + \frac{2}{3 - 2p}\right)\mathcal{L}\right)\mathbb{E}\left[f(x^k) - f(x^*)\right]$$
$$+\left(1 - \frac{p}{2}\right)\mathbb{E}\left[\psi^k\right]. \tag{45}$$

Next we would like to drop the second term in (45). For this we need to guarantee that

$$a_k - 2\alpha\mathcal{L}\left(b_k + \frac{2}{3 - 2p}\right) \geq 0 \tag{46}$$

Let $q \stackrel{\text{def}}{=} 1 - p$ so that the above becomes

$$\frac{q^{\frac{3k+2}{2}}(1-\sqrt{q})+1-q}{1-q^{\frac{3}{2}}} - 2\alpha\mathcal{L}\left(\frac{1+q^{2k+1}}{1+q}+\frac{2}{1+2q}\right) \geq 0.$$

In other words, after dividing through by $\left(\frac{1+q^{2k+1}}{1+q}+\frac{2}{1+2q}\right)$ and re-arranging, we require that

$$2\alpha\mathcal{L} \leq \frac{\frac{q^{\frac{3k+2}{2}}(1-\sqrt{q})+1-q}{1-q^{\frac{3}{2}}}}{\frac{1+q^{2k+1}}{1+q}+\frac{2}{1+2q}}$$

$$= \frac{\frac{q^{\frac{3k+2}{2}}(1-\sqrt{q})+1-q}{1-q^{\frac{3}{2}}}}{\frac{(1+q^{2k+1})(1+2q)+2(1+q)}{(1+q)(1+2q)}}$$

$$= \frac{q^{\frac{3k+2}{2}}(1-\sqrt{q})+1-q}{q^{2k+1}(1+2q)+3+4q}\frac{(1+q)(1+2q)}{1-q^{\frac{3}{2}}}. \tag{47}$$

We are now going to show that

$$\frac{q^{\frac{3k+2}{2}}(1-\sqrt{q})+1-q}{q^{2k+1}(1+2q)+3+4q} \geq \frac{1-q}{3+4q}. \tag{48}$$

Indeed, multiplying out the denominators of the above gives

$$F(q) \stackrel{\text{def}}{=} (3+4q)\left(q^{\frac{3k+2}{2}}(1-\sqrt{q})+1-q\right)-(1-q)\left(q^{2k+1}(1+2q)+3+4q\right)$$

$$= q^{\frac{3k+2}{2}}(1-\sqrt{q})(3+4q)-q^{2k+1}(1+2q)(1-q)$$

$$= q^{\frac{3k+2}{2}}(1-\sqrt{q})\left(3+4q-q^{\frac{k}{2}}(1+2q)(1+\sqrt{q})\right).$$

And since $q^{\frac{k}{2}} \leq 1$, we have

$$F(q) \geq q^{\frac{3k+2}{2}}(1-\sqrt{q})\left(3+4q-(1+2q)(1+\sqrt{q})\right)$$

$$= 2q^{\frac{3k+2}{2}}(1-\sqrt{q})(1-q\sqrt{q})$$

$$\geq 0.$$

As a result (48) holds. And thus if

$$2\alpha\mathcal{L} \leq \frac{1-q}{3+4q}\frac{(1+q)(1+2q)}{1-q^{\frac{3}{2}}}$$

holds, then (47) is verified for all $k$. This is why we impose the upper bound on the step size given in (20), which ensures that (46) is satisfied. Finally, this condition being verified, we get that

$$\mathbb{E}\left[\phi^{k+1}\right] \stackrel{(45)+(46)}{\leq} \left(1-\frac{2}{3}\alpha\mu\right)\mathbb{E}\left[\|x^k-x^*\|_2^2\right]+\left(1-\frac{p}{2}\right)\mathbb{E}\left[\psi^k\right]$$

$$\leq \beta\mathbb{E}\left[\phi^k\right], \tag{49}$$

where $\beta = \max\left\{1-\frac{2}{3}\alpha\mu, 1-\frac{p}{2}\right\}$. $\qquad\square$

## B.6 Proof of Corollary 5.1

*Proof.* We have that

$$\mathbb{E}\left[\phi^k\right] \leq \beta^k\phi^0,$$

where $\beta = \max\left\{1-\frac{1}{3\zeta_p}\frac{\mu}{\mathcal{L}(b)}, 1-\frac{p}{2}\right\}$. Hence using Lemma A.5, we have that the iteration complexity for an $\epsilon > 0$ approximate solution that verifies $\mathbb{E}\left[\phi^k\right] \leq \epsilon\phi^0$ is

$$2\max\left\{\frac{3\zeta_p}{2}\frac{\mathcal{L}(b)}{\mu}, \frac{1}{p}\right\}\log\left(\frac{1}{\epsilon}\right).$$

For the total complexity, one can notice that in expectation, we compute $2b+pn$ stochastic gradients at each iteration. $\qquad\square$

### B.7 Proof of Proposition 6.1

*Proof.* Dropping the $\log(1/\epsilon)$ for brevity, we distinguish two cases, $m \geq \frac{2(\mathcal{L}(b)+2\rho(b))}{\mu}$ and $m \leq \frac{2(\mathcal{L}(b)+2\rho(b))}{\mu}$.

1. $m \geq \frac{2(\mathcal{L}(b)+2\rho(b))}{\mu}$: Then $C_m(b) = 2(n+2bm)$, and hence we should use the smallest $m$ possible, that is, $m = \frac{2(\mathcal{L}(b)+2\rho(b))}{\mu}$.

2. $m \leq \frac{2(\mathcal{L}(b)+2\rho(b))}{\mu}$: Then $C_m(b) = \frac{2(n+2bm)}{m} \frac{2(\mathcal{L}(b)+2\rho(b))}{\mu} = 2\left(\frac{n}{m} + 2b\right) \frac{2(\mathcal{L}(b)+2\rho(b))}{\mu}$. Hence $C_m(b)$ is decreasing in $m$ and we should then use the highest possible value for $m$, that is $m = \frac{2(\mathcal{L}(b)+2\rho(b))}{\mu}$.

The result now follows by substituting $m = \frac{2(\mathcal{L}(b)+2\rho(b))}{\mu}$ into (18). $\qquad\square$

### B.8 Proof of Proposition 6.2

*Proof.* Recall that have from Lemma 4.3:

$$\mathcal{L}(b) = \frac{1}{b}\frac{n-b}{n-1}L_{\max} + \frac{n}{b}\frac{b-1}{n-1}L, \tag{50}$$

$$\rho(b) = \frac{1}{b}\frac{n-b}{n-1}L_{\max}. \tag{51}$$

For brevity, we temporarily drop the term $\log\left(\frac{1}{\epsilon}\right)$ in $C_m(b)$ defined in Equation (18). Hence, we want to find, for different values of $m$:

$$b^* = \arg\min_{b\in[n]} C_m(b) := 2\left(\frac{n}{m} + 2b\right)\max\{\kappa(b), m\}, \tag{52}$$

where $\kappa(b) \overset{\text{def}}{=} \frac{\mathcal{L}(b)+2\rho(b)}{\mu}$.

**When $m = n$.** In this case we have

$$C_n(b) \overset{(18)}{=} 2(2b+1)\max\{\kappa(b), n\}, \tag{53}$$

Writing $\kappa(b)$ explicitly:

$$\kappa(b) = \frac{1}{\mu(n-1)}\left((3L_{\max} - L)\frac{n}{b} + nL - 3L_{\max}\right).$$

Since $3L_{\max} > L$, $\kappa(b)$ is a decreasing function of $b$. In the light of this observation, we will determine the optimal mini-batch size. The upcoming analysis is summarized in Table 1.

We distinguish three cases:

- If $n \leq \frac{L}{\mu}$: then $\kappa(n) = \frac{L}{\mu} \geq n$. Since $\kappa(b)$ is decreasing, this means that for all $b \in [n], \kappa(b) \geq n$. Consequently, $C_n(b) = 2(2b+1)\kappa(b)$. Differentiating twice:

$$C_n''(b) = \frac{4}{\mu(n-1)}\frac{(3L_{\max}-L)n}{b^3} > 0.$$

Hence $C_n(b)$ is a convex function. Now examining its first derivative:

$$C_n'(b) = \frac{2}{\mu(n-1)}\left(-\frac{(3L_{\max}-L)n}{b^2} + 2(nL - 3L_{\max})\right),$$

we can see that:

- If $n \leq \frac{3L_{\max}}{L}$, $C_n(b)$ is a decreasing function, hence

$$b^* = n.$$

- If $n > \frac{3L_{\max}}{L}$, $C_n(b)$ admits a minimizer, which we can find by setting its first derivative to zero. The solution is

$$\hat{b} \overset{\text{def}}{=} \sqrt{\frac{n}{2} \frac{3L_{\max} - L}{nL - 3L_{\max}}}.$$

Hence,

$$b^* = \left\lfloor \hat{b} \right\rfloor$$

- If $n \geq \frac{3L_{\max}}{\mu}$, then $\kappa(1) = 3\frac{L_{\max}}{\mu}$. Since $\kappa(b)$ is decreasing, this means that for all $b \in [n], \kappa(b) \leq n$. Hence, $C_n(b) = 2(2b+1)n$. $C_n(b)$ is an increasing function of $b$. Therefore,

$$b^* = 1.$$

- If $\frac{L}{\mu} < n < \frac{3L_{\max}}{\mu}$, we have $\kappa(1) > n$ and $\kappa(n) < n$. Hence there exists $\tilde{b} \in [1, n]$ such that $\kappa(b) = n$, and it is given by

$$\tilde{b} \overset{\text{def}}{=} \frac{(3L_{\max} - L)n}{n(n-1)\mu - nL + 3L_{\max}}. \tag{54}$$

Define $G(b) \overset{\text{def}}{:=} (2b+1)\kappa(b)$. Then,

$$\underset{b \in [1, n]}{\arg\min} \, G(b) = \begin{cases} n & \text{if } n \leq \frac{3L_{\max}}{L}, \\ \hat{b} & \text{if } n > \frac{3L_{\max}}{L}. \end{cases} \tag{55}$$

As a result, we have that

- if $n \leq \frac{3L_{\max}}{L}$, $G(b)$ is decreasing on $[1, n]$, hence $C_n(b)$ is decreasing on $[1, \tilde{b}]$ and increasing on $[\tilde{b}, n]$. Then,

$$b^* = \left\lfloor \tilde{b} \right\rfloor.$$

- if $n > \frac{3L_{\max}}{L}$, $G(b)$ is decreasing on $[1, \hat{b}]$ and increasing on $[\hat{b}, n]$. Hence $C_n(b)$ is decreasing on $[1, \min\{\hat{b}, \tilde{b}\}]$ and increasing on $[\min\{\hat{b}, \tilde{b}\}, 1]$. Then,

$$b^* = \left\lfloor \min\{\hat{b}, \tilde{b}\} \right\rfloor.$$

To summarize, we have for $m = n$,

$$b^* = \begin{cases} 1 & \text{if } n \geq \frac{3L_{\max}}{\mu} \\ \left\lfloor \min(\tilde{b}, \hat{b}) \right\rfloor & \text{if } \max\{\frac{L}{\mu}, \frac{3L_{\max}}{L}\} < n < \frac{3L_{\max}}{\mu} \\ \left\lfloor \hat{b} \right\rfloor & \text{if } \frac{3L_{\max}}{L} < n < \frac{L}{\mu} \\ \left\lfloor \tilde{b} \right\rfloor & \text{if } \frac{L}{\mu} < n \leq \frac{3L_{\max}}{L} \\ n & \text{otherwise, if } n \leq \min\{\frac{L}{\mu}, \frac{3L_{\max}}{L}\} \end{cases} \tag{56}$$

**When $m = n/b$.** In this case we have

$$C_m(b) \overset{(18)}{=} 6 \max\{b\kappa(b), n\},$$

with

$$b\kappa(b) = \frac{1}{\mu(n-1)} \left( (3L_{\max} - L)n + (nL - 3L_{\max})b \right),$$

and thus $\kappa(1) = \frac{3L_{\max}}{\mu}$ and $n\kappa(n) = \frac{nL}{\mu} \geq n$. We distinguish two cases:

- if $n \leq \frac{3L_{\max}}{L}$, then $b\kappa(b)$ is decreasing in $b$. Since $n\kappa(n) \geq n$, $C_m(b) = 6b\kappa(b)$, thus $C_m(b)$ is decreasing in $b$, hence

$$b^* = n$$

- if $n > \frac{3L_{\max}}{L}$, $b\kappa(b)$ is increasing in $b$. Thus,

  – if $n \leq \frac{3L_{\max}}{\mu} = \kappa(1)$, then $C_m(b) = 6b\kappa(b)$. Hence $b^* = 1$.

  – if $n > \frac{3L_{\max}}{\mu}$, using the definition of $\tilde{b}$ in Equation (54), we have that

  $$C_m(b) = \begin{cases} 6n & \text{for } b \in [1, \bar{b}] \\ 6b\kappa(b) & \text{for } b \in [\bar{b}, n] \end{cases},$$

  where

  $$\bar{b} = \frac{n(n-1)\mu - (3L_{\max} - L)n}{nL - 3L_{\max}}$$

  is the batch size $b \in [n]$ which verifies $b\kappa(b) = n$. Hence $b^*$ can be any point in $\{1, \ldots, \lfloor \bar{b} \rfloor\}$. In light of shared memory parallelism, $b^* = \lfloor \bar{b} \rfloor$ would be the most practical choice.

  $\square$

## C  Optimal mini-batch size for Algorithm 2

By using a similar proof as in Section B.8, we derive the following result.

**Proposition C.1.** *Note $b^* \overset{def}{=} \underset{b \in [n]}{\arg\min}\, C_p(b)$, where $C_p(b)$ is defined in (22). For the widely used choice $p = \frac{1}{n}$, we have that*

$$b^* = \begin{cases} 1 & \text{if } n \geq \frac{3\zeta_{1/n}}{2} \frac{L_{\max}}{\mu} \\ \left\lfloor \min(\tilde{b}, \hat{b}) \right\rfloor & \text{if } \frac{3\zeta_{1/n}}{2} \frac{L}{\mu} < n < \frac{3\zeta_{1/n}}{2} \frac{L_{\max}}{\mu} \\ \left\lfloor \hat{b} \right\rfloor & \text{otherwise, if } n \leq \frac{3\zeta_{1/n}}{2} \frac{L}{\mu} \end{cases}, \tag{57}$$

*where $\zeta_p$ is defined in (20) for $p \in (0,1]$ and:*

$$\hat{b} = \sqrt{\frac{n}{2} \frac{L_{\max} - L}{nL - L_{\max}}}, \quad \tilde{b} = \frac{\frac{3\zeta_p}{2} n(L_{\max} - L)}{\mu n(n-1) - \frac{3\zeta_p}{2}(nL - L_{\max})}.$$

Because $\zeta_p$ depends on $p$, optimizing the total complexity with respect to $b$ for the case $p = \frac{b}{n}$ is extremely cumbersome. Thus, we restrain our study for the optimal mini-batch sizes for Algorithm 2 to the case where $p = \frac{1}{n}$.

*Proof.* For brevity, we temporarily drop the term $\log\left(\frac{1}{\epsilon}\right)$ in $C_p(b)$ defined in Equation (22). Hence, we want to find, for different values of $m$:

$$b^* = \underset{b \in [n]}{\arg\min}\, C_{1/n}(b) := 2(2b+1)\max\{\pi(b), m\}, \tag{58}$$

where $\pi(b) \overset{def}{=} \frac{3\zeta_p}{2} \frac{\mathcal{L}(b)}{\mu}$. We have

$$\pi(b) = \frac{3\zeta_p}{2} \frac{1}{\mu(n-1)} \left( \frac{n(L_{\max} - L)}{b} + nL - L_{\max} \right). \tag{59}$$

Since $L_{\max} \geq L$, $\pi(b)$ is a decreasing function on $[1, n]$. We distinguish three cases:

- if $n > \pi(1) = \frac{3\zeta_p}{2} \frac{L_{\max}}{\mu}$, then for all $b \in [1, n], n > \pi(b)$. Hence,

  $$C_n(b) = 2(2b+1)n.$$

  $C_{1/n}(b)$ is an increasing function of $b$. Hence

  $$b^* = 1.$$

- if $n < \pi(n) = \frac{3\zeta_p}{2}\frac{L}{\mu}$, then for all $b \in [1, n]$, $n < \pi(b)$. Hence,

$$C_{1/n}(b) = 2(2b + 1)\pi(b).$$

Now, consider the function

$$
\begin{aligned}
G(b) \quad &\overset{\text{def}}{=} \quad (2b + 1)\pi(b) \\
&= \quad \frac{3\zeta_p}{2}\frac{1}{\mu(n - 1)}\left(2(nL - L_{\max})b + \frac{n(L_{\max} - L)}{b}\right) + \Omega,
\end{aligned}
$$

where $\Omega$ replaces constants which don't depend on $b$. The first derivative of $G(b)$ is

$$G'(b) = \frac{3\zeta_p}{2}\frac{1}{\mu(n - 1)}\left(-\frac{n(L_{\max} - L)}{b^2} + 2(nL - L_{\max})\right),$$

and its second derivative is

$$G''(b) = \frac{3\zeta_p n(L_{\max} - L)}{\mu(n - 1)b^3} \geq 0.$$

$G(b)$ is a convex function, and we can find its minimizer by setting its first derivative to zero. This minimizer is

$$\hat{b} \overset{\text{def}}{=} \sqrt{\frac{n}{2}\frac{L_{\max} - L}{nL - L_{\max}}}.$$

Indeed, recall that from Lemma A.6, we have $nL \geq L_{\max}$.

Thus, in this case, $C_{1/n}(b)$ is a convex function and its minimizer os

$$b^* = \left\lfloor \hat{b} \right\rfloor.$$

- if $\frac{3\zeta_p}{2}\frac{L}{\mu} = \pi(n) \leq n \leq \pi(1) = \frac{3\zeta_p}{2}\frac{L_{\max}}{\mu}$. Then there exists $b \in [1, n]$ such that $\pi(b) = n$ and its expression is given by

$$\tilde{b} = \frac{\frac{3\zeta_p}{2}n(L_{\max} - L)}{\mu n(n - 1) - \frac{3\zeta_p}{2}(nL - L_{\max})}.$$

Consequently, the function $C_n(b)$ is decreasing on $\left[1, \min\left\{\tilde{b}, \hat{b}\right\}\right]$ and increasing on $\left[\min\left\{\tilde{b}, \hat{b}\right\}, n\right]$. Hence,

$$b^* = \left\lfloor \min\left\{\tilde{b}, \hat{b}\right\} \right\rfloor.$$

$\square$

## D  Samplings

In Definition 3.3, we defined $b$–nice sampling. For completeness, we present here some other interesting possible samplings.

**Definition D.1** (single-element sampling). *Given a set of probabilities $(p_i)_{i \in [n]}$, $S$ is a single-element sampling if $\mathbb{P}(|S| = 1) = 1$ and*

$$\mathbb{P}(S = \{i\}) = p_i \quad \forall i \in [n].$$

**Definition D.2** (partition sampling). *Given a partition $\mathcal{B}$ of $[n]$, $S$ is a partition sampling if*

$$p_B \overset{\text{def}}{=} \mathbb{P}(S = B) > 0 \quad \forall B \in \mathcal{B}, \text{ and } \sum_{B \in \mathcal{B}} p_B = 1.$$

**Definition D.3** (independent sampling). *$S$ is an independent sampling if it includes every $i$ independently with probability $p_i > 0$.*

In Section E, we will determine for each of these samplings their corresponding expected smoothness constant.

# E  Expected Smoothness

First, we present two general properties about the expected smoothness constant presented in Lemma 4.1: we establish its existence, and we prove that it is always greater than the strong convexity constant. Then, we determine the expected smoothness constant for particular samplings.

## E.1  General properties of the expected smoothness constant

The following lemma is an adaptation of Theorem 3.6 in [8]. It establishes the existence of the expected smoothness constant as a result of the smoothness and convexity of the functions $f_i, i \in [n]$.

**Lemma E.1** (Theorem 3.6 in [8]). *Let $v$ be a sampling vector as defined in Definition 3.1 with $v_i \geq 0$ with probability one . Suppose that $f_v(w) = \frac{1}{n} \sum_{i=1}^{n} f_i(w)v_i$ is $L_v$–smooth and convex. It follows that the expected smoothness constant* (4.1) *is given by*

$$\mathcal{L} = \max_{i \in [n]} \mathbb{E}\left[L_v v_i\right].$$

*Proof.* Since the $f_i$'s are convex, each realization of $f_v$ is convex, and it follows from equation 2.1.7 in [19] that

$$\|\nabla f_v(x) - \nabla f_v(y)\|_2^2 \quad \leq \quad 2L_v \left(f_v(x) - f_v(y) - \langle \nabla f_v(y), x - y \rangle\right). \tag{60}$$

Taking expectation over the sampling gives

$$
\begin{aligned}
\mathbb{E}\left[\|\nabla f_v(x) - \nabla f_v(x^*)\|_2^2\right] &\overset{(60)}{\leq} 2\mathbb{E}\left[L_v \left(f_v(x) - f_v(x^*) - \langle \nabla f_v(x^*), x - x^* \rangle\right)\right] \\
&\overset{(7)}{=} \frac{2}{n}\mathbb{E}\left[\sum_{i=1}^{n} L_v v_i \left(f_i(x) - f_i(x^*) - \langle \nabla f_i(x^*), x - x^* \rangle\right)\right] \\
&= \frac{2}{n}\sum_{i=1}^{n} \mathbb{E}\left[L_v v_i\right] \left(f_i(x) - f_i(y) - \langle \nabla f_i(x^*), x - x^* \rangle\right) \\
&\overset{(1)}{\leq} 2 \max_{i=1,\ldots,n} \mathbb{E}\left[L_v v_i\right] \left(f(x) - f(x^*) - \langle \nabla f(x^*), x - x^* \rangle\right) \\
&= 2 \max_{i=1,\ldots,n} \mathbb{E}\left[L_v v_i\right] \left(f(x) - f(x^*)\right).
\end{aligned}
$$

By comparing the above with (11) we have that $\mathcal{L} = \max_{i=1,\ldots,n} \mathbb{E}\left[L_v v_i\right]$. $\qquad \square$

**Lemma E.2** (PL inequality). *If $f$ is $\mu$–strongly convex, then for all $x, y \in \mathbb{R}^d$*

$$\frac{1}{2\mu}\|\nabla f(x)\|_2^2 \geq f(x) - f(x^*), \quad \forall x \in \mathbb{R}^d. \tag{61}$$

*Proof.* Since $f$ is $\mu$–strongly convex, we have from, rearranging (6), that for all $x, y \in \mathbb{R}^d$

$$f(y) \geq f(x) + \langle \nabla f(x), y - x \rangle + \frac{\mu}{2}\|x - y\|_2^2.$$

Minimizing both sides of this inequality in $y$ proves (61). $\qquad \square$

The following lemma shows that the expected smoothness constant is always greater than the strong convexity constant.

**Lemma E.3.** *If the expected smoothness inequality* (11) *holds with constant $\mathcal{L}$ and $f$ is $\mu$–strongly convex, then $\mathcal{L} \geq \mu$.*

*Proof.* We have, since $\mathbb{E}\left[\nabla f_v(x) - \nabla f_v(x^*)\right] = \nabla f(x)$

$$
\begin{aligned}
\mathbb{E}\left[\|\nabla f_v(x) - \nabla f_v(x^*) - \nabla f(x)\|_2^2\right] &\overset{\text{Lem. }A.2}{=} \mathbb{E}\left[\|\nabla f_v(x) - \nabla f_v(x^*)\|_2^2\right] - \|\nabla f(x)\|_2^2 \\
&\overset{(11)+(61)}{\leq} 2(\mathcal{L} - \mu)(f(x) - f(x^*)). \tag{62}
\end{aligned}
$$

Hence $2(\mathcal{L} - \mu)(f(x) - f(x^*)) \geq 0$, which means $\mathcal{L} \geq \mu$. $\qquad \square$

**Remark E.1.** *Consider the expected residual constant $\rho$ defined in 4.2. This constant verifies for all $x \in \mathbb{R}^d$,*

$$\mathbb{E}\left[\|\nabla f_v(x) - \nabla f_v(x^*) - \nabla f(x)\|\right] \leq 2\rho(f(x) - f(x^*)).$$

*From Equation* (62)*, we can see that we can use $\rho = \mathcal{L}$ as the expected residual constant.*

### E.2 Expected smoothness constant for particular samplings

The results on the expected smoothness constants related to the samplings we present here are all derived in [8] and thus are given without proof. The expected smoothness constant for $b$-nice sampling is given in Lemma 4.3. Here, we present this constant for single-element sampling, partition sampling and independent sampling.

**Lemma E.4** ($\mathcal{L}$ for single-element sampling. Proposition 3.7 in [8])**.** *Consider $S$ a single-element sampling from Definition D.1. If for all $i \in [n]$, $f_i$ is $L_i$–smooth, then*

$$\mathcal{L} = \frac{1}{n} \max_{i \in [n]} \frac{L_i}{p_i}$$

*where $p_i = \mathbb{P}(S = \{i\})$.*

**Remark E.2.** *Consider $S$ a single-element sampling from Definition D.1. Then, the probabilities that maximize $\mathcal{L}$ are*

$$p_i = \frac{L_i}{\sum_{j \in [n]} L_j}.$$

*Consequently,*

$$\mathcal{L} = \bar{L} \overset{def}{=} \frac{1}{n} \sum_{i=1}^{n} L_i.$$

In contrast, for uniform single-element sampling, *i.e.*, when $p_i = \frac{1}{n}$ for all $i$, we have $\mathcal{L} = L_{\max}$, which can be significantly larger than $\bar{L}$. Since the step sizes of all our algorithms are a decreasing function of $\mathcal{L}$, importance sampling can lead to much faster algorithms.

**Lemma E.5** ($\mathcal{L}$ for partition sampling. Proposition 3.7 in [8])**.** *Given a partition $\mathcal{B}$ of $[n]$, consider $S$ a partition sampling from Definition D.3. For all $B \in \mathcal{B}$, suppose that $f_B(x) \overset{def}{=} \frac{1}{b} \sum_{i \in B} f_i(x)$ is $L_B$–smooth. Then, with $p_B = \mathbb{P}(S = B)$*

$$\mathcal{L} = \frac{1}{n} \max_{B \in \mathcal{B}} \frac{L_B}{p_B}$$

**Lemma E.6** ($\mathcal{L}$ for independent sampling. Proposition 3.8 in [8])**.** *Consider $S$ a single-element sampling from Definition D.3. Note $p_i = \mathbb{P}(i \in S)$. If for all $i \in [n]$, $f_i$ is $L_i$–smooth and $f$ is $L$–smooth, then*

$$\mathcal{L} = L + \max_{i \in [n]} \frac{1 - p_i}{p_i} \frac{L_i}{n}$$

*where $p_i = \mathbb{P}(S = \{i\})$.*

## F Expected residual

In this section, we compute bounds on the expected residual $\rho$ from Lemma 4.2.

**Lemma F.1.** *Let $v = [v_1, \ldots, v_n] \in \mathbb{R}^n$ be an unbiased sampling vector with $v_i \geq 0$ with probability one. It follows that the expected residual constant exists with*

$$\rho = \frac{\lambda_{\max}(\text{Var}\,[v])}{n} L_{\max}, \tag{63}$$

*where $\text{Var}\,[v] = \mathbb{E}\left[(v - \mathbb{1})(v - \mathbb{1})^\top\right].$*

Before the proof, let us introduce the following lemma (inspired from `https://www.cs.ubc.ca/~nickhar/W12/NotesMatrices.pdf`).

**Lemma F.2** (Trace inequality). *Let $A$ and $B$ be symmetric $n \times n$ such that $A \succcurlyeq 0$. Then,*

$$Tr\,(AB) \leq \lambda_{\max}(B)Tr\,(A)$$

*Proof.* Let $A = \sum_{i=1}^{n} \lambda_i(A)U_iU_i^\top$, where $\lambda_1(A) \geq \ldots \geq \lambda_n(A) \geq 0$ denote the ordered eigenvalues of matrix $A$. Setting $V_i \stackrel{\text{def}}{=} \sqrt{\lambda_i(A)}U_i$ for all $i \in [n]$, we can write $A = \sum_{i=1}^{n} V_iV_i^\top$. Then,

$$\mathrm{Tr}\,(AB) = \mathrm{Tr}\left(\sum_{i=1}^{n} V_iV_i^\top B\right) = \sum_{i=1}^{n} \mathrm{Tr}\left(V_iV_i^\top B\right) = \sum_{i=1}^{n} \mathrm{Tr}\left(V_i^\top BV_i\right) = \sum_{i=1}^{n} V_i^\top BV_i$$

$$\leq \lambda_{\max}(B)\sum_{i=1}^{n} V_i^\top V_i = \lambda_{\max}(B)\mathrm{Tr}\,(A)\,,$$

where we use in the inequality that $B \preccurlyeq \lambda_{\max}(B)I_n$. $\qquad\square$

We now turn to the proof of the theorem.

*Proof.* Let $v = [v_1, \ldots, v_n] \in \mathbb{R}^n$ be an unbiased sampling vector with $v_i \geq 0$ with probability one. We will show that there exists $\rho \in \mathbb{R}_+$ such that:

$$\mathbb{E}\left[\|\nabla f_v(w) - \nabla f_v(x^*) - (\nabla f(w) - \nabla f(x^*))\|_2^2\right] \leq 2\rho\,(f(w) - f(x^*))\,. \qquad (64)$$

Let us expand the squared norm first. Define $DF(w)$ as the Jacobian of $F(w) \stackrel{def}{=} [f_1(w), \ldots, f_n(w)]$ We denote $R \stackrel{\text{def}}{=} (DF(w) - DF(x^*))$

$$C \stackrel{\text{def}}{=} \|\nabla f_v(w) - \nabla f_v(x^*) - (\nabla f(w) - \nabla f(x^*))\|_2^2$$

$$= \frac{1}{n^2}\|(DF(w) - DF(x^*))(v - \mathbb{1})\|_2^2$$

$$= \frac{1}{n^2}\langle R(v - \mathbb{1}), R(v - \mathbb{1})\rangle_{\mathbb{R}^d}$$

$$= \frac{1}{n^2}\mathrm{Tr}\left((v - \mathbb{1})^\top R^\top R(v - \mathbb{1})\right)$$

$$= \frac{1}{n^2}\mathrm{Tr}\left(R^\top R(v - \mathbb{1})(v - \mathbb{1})^\top\right).$$

Taking expectation,

$$\mathbb{E}\,[C] = \frac{1}{n^2}\mathrm{Tr}\left(R^\top R\mathrm{Var}\,[v]\right)$$

$$\leq \frac{1}{n^2}\mathrm{Tr}\left(R^\top R\right)\lambda_{\max}(\mathrm{Var}\,[v]). \qquad (65)$$

Moreover, since the $f_i$'s are convex and $L_i$-smooth, it follows from equation 2.1.7 in [19] that

$$\mathrm{Tr}\left(R^\top R\right) = \sum_{i=1}^{n}\|\nabla f_i(w) - \nabla f_i(x^*)\|_2^2$$

$$\leq 2\sum_{i=1}^{n} L_i(f_i(w) - f_i(x^*) - \langle\nabla f_i(x^*), w - x^*\rangle)$$

$$\leq 2nL_{\max}(f(w) - f(x^*)). \qquad (66)$$

Therefore,

$$\mathbb{E}[C] \overset{(65)+(66)}{\leq} 2\frac{\lambda_{\max}(\mathrm{Var}\,[v])}{n}L_{\max}(f(w) - f(x^*)). \tag{67}$$

Which means

$$\rho = \frac{\lambda_{\max}(\mathrm{Var}\,[v])}{n}L_{\max} \tag{68}$$

$\square$

Hence depending on the sampling $S$, we need to study the eigenvalues of the matrix $\mathrm{Var}\,[v]$, whose general term is given by

$$(\mathrm{Var}\,[v])_{ij} = \begin{cases} \frac{1}{p_i} - 1 & \text{if } i = j \\ \frac{P_{ij}}{p_i p_j} - 1 & \text{otherwise,} \end{cases} \tag{69}$$

with

$$p_i \overset{\text{def}}{=} \mathbb{P}(i \in S) \text{ and } P_{ij} \overset{\text{def}}{=} \mathbb{P}(i \in S, j \in S) \text{ for } i, j \in [n] \tag{70}$$

To specialize our results to particular samplings, we introduce some notations:

- $\mathcal{B}$ designates all the possible sets for the sampling $S$,
- $b = |B|$, where $B \in \mathcal{B}$, when the sizes of all the elements of $\mathcal{B}$ are equal.

## F.1 Expected residual for uniform $b$-nice sampling

**Lemma F.3** ($\rho$ for $b$-nice sampling). *Consider $b$-nice sampling from Definition 3.3. If each $f_i$ is $L_{\max}$-smooth, then*

$$\rho = \frac{n - b}{(n - 1)b}L_{\max}. \tag{71}$$

*Proof.* For uniform $b$-nice sampling, we have using notations from (70)

$$\forall i \in [n], p_i = \frac{c_1}{|\mathcal{B}|},$$

$$\forall i, j \in [n], P_{ij} = \frac{c_2}{|\mathcal{B}|},$$

with $c_1 = \binom{n-1}{b-1}$, $c_2 = \binom{n-2}{b-2}$ and $|\mathcal{B}| = \binom{n}{b}$. Hence,

$$\mathrm{Var}\,[v] \overset{(69)}{=} \begin{bmatrix} \frac{|\mathcal{B}|}{c_1} - 1 & \frac{|\mathcal{B}|c_2}{c_1^2} - 1 & \cdots & \frac{|\mathcal{B}|c_2}{c_1^2} - 1 & \frac{|\mathcal{B}|c_2}{c_1^2} - 1 \\ \frac{|\mathcal{B}|c_2}{c_1^2} - 1 & \frac{|\mathcal{B}|}{c_1} - 1 & \cdots & \frac{|\mathcal{B}|c_2}{c_1^2} - 1 & \frac{|\mathcal{B}|c_2}{c_1^2} - 1 \\ \vdots & & \ddots & & \vdots \\ \frac{|\mathcal{B}|c_2}{c_1^2} - 1 & \cdots & \cdots & \frac{|\mathcal{B}|}{c_1} - 1 & \frac{|\mathcal{B}|c_2}{c_1^2} - 1 \\ \frac{|\mathcal{B}|c_2}{c_1^2} - 1 & \cdots & \cdots & \frac{|\mathcal{B}|c_2}{c_1^2} - 1 & \frac{|\mathcal{B}|}{c_1} - 1 \end{bmatrix}.$$

As noted in Appendix C of [9], $\mathrm{Var}\,[v]$ is then a circulant matrix with associated vector

$$\left(\frac{|\mathcal{B}|}{c_1} - 1, \frac{|\mathcal{B}|c_2}{c_1^2} - 1, \ldots, \frac{|\mathcal{B}|c_2}{c_1^2} - 1\right),$$

and, as such, it has two eigenvalues

$$\lambda_1 \overset{\text{def}}{=} \frac{|\mathcal{B}|}{c_1}\left(1 + (n-1)\frac{c_2}{c_1}\right) - n = 0,$$

$$\lambda_2 \overset{\text{def}}{=} \frac{|\mathcal{B}|}{c_1}\left(1 - \frac{c_2}{c_1}\right) = \frac{n(n-b)}{b(n-1)}. \tag{72}$$

Hence, the expected residual can be computed explicitly as

$$\rho \overset{(68)}{=} \frac{n-b}{(n-1)b} L_{\max}.$$
(73)

$\square$

We can see that the residual constant is a decreasing function of $b$ and in particular: $\rho(1) = L_{\max}$ and $\rho(n) = 0$.

### F.2 Expected residual for uniform partition sampling

**Lemma F.4** ($\rho$ for uniform partition sampling). *Suppose that $b$ divises $n$ and consider partition sampling from Definition D.2. Given a partition $\mathcal{B}$ of $[n]$ of size $\frac{b}{n}$, if each $f_i$ is $L_{\max}$-smooth, then,*

$$\rho = \left(1 - \frac{b}{n}\right) L_{\max}.$$
(74)

*Proof.* Recall that for partition sampling, we choose *a priori* a partition $\mathcal{B} = B_1 \sqcup \cdots \sqcup B_{\frac{n}{b}}$ of $[n]$. Then, for $k \in [\frac{n}{b}]$,

$$\forall i \in [n], p_i = \begin{cases} p_{B_k} = \frac{b}{n} & \text{if } i \in B_k \\ 0 & \text{otherwise,} \end{cases}$$
(75)

$$\forall i, j \in [n], P_{ij} = \begin{cases} p_{B_k} = \frac{b}{n} & \text{if } i, j \in B_k \\ 0 & \text{otherwise.} \end{cases}$$
(76)

Let $k \in [\frac{n}{b}]$. If $i, j \in B_k$, then $\frac{1}{p_i} - 1 = \frac{P_{ij}}{p_i p_j} - 1 = \frac{n}{b} - 1$.

As a result, up to a reordering of the observations, $\text{Var}[v]$ is a block diagonal matrix, whose diagonal matrices, which are all equal, are given by, for $k \in [\frac{n}{b}]$,

$$V_k = (\frac{n}{b} - 1)\mathbb{1}_b \mathbb{1}_b^\top = \begin{bmatrix} \frac{n}{b} - 1 & \frac{n}{b} - 1 & \cdots & \frac{n}{b} - 1 & \frac{n}{b} - 1 \\ \frac{n}{b} - 1 & \frac{n}{b} - 1 & \cdots & \frac{n}{b} - 1 & \frac{n}{b} - 1 \\ \vdots & & \ddots & & \vdots \\ \frac{n}{b} - 1 & \cdots & \cdots & \frac{n}{b} - 1 & \frac{n}{b} - 1 \\ \frac{n}{b} - 1 & \cdots & \cdots & \frac{n}{b} - 1 & \frac{n}{b} - 1 \end{bmatrix} \in \mathbb{R}^{b \times b}.$$

Since all the matrices on the diagonal are equal, the eigenvalues of $\text{Var}[v]$ are simply those of one of these matrices. Any matrix $V_k = (\frac{n}{b} - 1)\mathbb{1}_b \mathbb{1}_b^\top$ we consider has two eigenvalues: $0$ and $n - b$. Then,

$$\rho \overset{(68)}{=} \left(1 - \frac{b}{n}\right) L_{\max}.$$
(77)

$\square$

If $b = n$, SVRG with uniform partition sampling boils down to gradient descent as we recover $\rho = 0$. For $b = 1$, we have $\rho = \left(1 - \frac{1}{n}\right) L_{\max}$.

### F.3 Expected residual for independent sampling

**Lemma F.5** ($\rho$ for independent sampling). *Consider independent sampling from Definition D.2. Let $p_i = \mathbb{P}(i \in S)$. If each $f_i$ is $L_{\max}$-smooth, then*

$$\rho = \left(\frac{1}{\min\limits_{i \in [n]} p_i} - 1\right) \frac{L_{\max}}{n}.$$
(78)

*Proof.* Using the notations from (70), we have

$$\forall i \in [n], p_i \;=\; p_i,$$
$$\forall i,j \in [n], P_{ij} \;=\; p_i p_j \quad \text{when } i \neq j.$$

Thus, according to (69):

$$\text{Var}\,[v] = \text{Diag}\left(\frac{1}{p_1}-1, \frac{1}{p_2}-1, \ldots, \frac{1}{p_n}-1\right).$$

whose largest eigenvalue is

$$\lambda_{\max}(\text{Var}\,[v]) = \max_{i \in [n]} \frac{1}{p_i} - 1 = \frac{1}{\min\limits_{i \in [n]} p_i} - 1.$$

Consequently,

$$\rho \overset{(68)}{=} \left(\frac{1}{\min\limits_{i \in [n]} p_i} - 1\right) \frac{L_{\max}}{n}. \tag{79}$$

$\square$

If $p_i = \frac{1}{n}$ for all $i \in [n]$, which corresponds in expectation to uniform single-element sampling SVRG since $\mathbb{E}\,[|S|] = 1$, we have $\rho = \frac{n-1}{n} L_{\max}$. While if $p_i = 1$ for all $i \in [n]$, this leads to gradient descent and we recover $\rho = 0$.

The following remark gives a condition to construct an independent sampling with $E|S| = b$.

**Remark F.1.** *One can add the following condition on the probabilities:* $\sum_{i=1}^{n} p_i = b$, *such that* $\mathbb{E}\,[|S|] = b$. *Such a sampling is called b-independent sampling. This condition is obviously met if* $p_i = \frac{b}{n}$ *for all* $i \in [n]$.

**Lemma F.6.** *Let $S$ be a independent sampling from $[n]$ and let $p_i = \mathbb{P}\,[i \in S]$ for all $i \in [n]$. If* $\sum_{i=1}^{n} p_i = b$, *then* $\mathbb{E}\,[|S|] = b$.

*Proof.* Let us model our sampling by a tossing of $n$ independent rigged coins. Let $X_1, \ldots, X_n$ be $n$ Bernoulli random variables representing these tossed coin, *i.e.*, $X_i \sim \mathcal{B}(p_i)$, with $p_i \in [0,1]$ for $i \in [n]$. If $X_i = 1$, then the point $i$ is selected in the sampling $S$. Thus the number of selected points in the mini-batch $|S|$ can be denoted as the following random variable $\sum_{i=1}^{n} X_i$, and its expectation equals

$$\mathbb{E}\,[|S|] = \mathbb{E}\left[\sum_{i=1}^{n} X_i\right] = \sum_{i=1}^{n} \mathbb{E}\,[X_i] = \sum_{i=1}^{n} p_i = b \;\;.$$

$\square$

**Remark F.2.** *Note that one does not need the independence of the* $(X_i)_{i=1,\ldots,n}$.

### F.4 Expected residual for single-element sampling

From Remark E.1, we can take $\mathcal{L}$ as the expected residual constant. Thus, we simply use the expected smoothness constant from Lemma E.4.

**Lemma F.7** ($\rho$ for single-element sampling)**.** *Consider single-element sampling from Definition D.1. If for all $i \in [n]$, $f_i$ is $L_i$-smooth, then*

$$\rho = \frac{1}{n} \max_{i \in [n]} \frac{L_i}{p_i}.$$

## G   Additional experiments

### G.1   Comparison of theoretical variants of SVRG

In this series of experiments, we compare the performance of the SVRG algorithm with the settings of [13] against *Free-SVRG* and *L-SVRG-D* with the settings given by our theory.

### G.1.1 Experiment 1.a: comparison without mini-batching ($b = 1$)

A widely used choice for the size of the inner loop is $m = n$. Since our algorithms allow for a free choice of the size of the inner loop, we set $m = n$ for *Free-SVRG* and $p = 1/n$ for *L-SVRG-D*, and use a mini-batch size $b = 1$. For vanilla *SVRG*, we set $m$ to its theoretical value $20L_{\max}/\mu$ as in [4]. See Figures 4, 5, 6 and 7. We can see that *Free-SVRG* and *L-SVRG-D* often outperform the SVRG algorithm [13]. It is worth noting that, in Figure 4a, 6a and 7 the classic version of SVRG can lead to increase of the suboptimality when entering the outer loop. This is due to the fact that the reference point is set to a weighted average of the iterates of the inner loop, instead of the last iterate.

(a) $\lambda = 10^{-1}$

(b) $\lambda = 10^{-3}$

Figure 4: Comparison of theoretical variants of SVRG without mini-batching ($b = 1$) on the *YearPredictionMSD* data set.

(a) $\lambda = 10^{-1}$

(b) $\lambda = 10^{-3}$

Figure 5: Comparison of theoretical variants of SVRG without mini-batching ($b = 1$) on the *slice* data set.

(a) $\lambda = 10^{-1}$

(b) $\lambda = 10^{-3}$

Figure 6: Comparison of theoretical variants of SVRG without mini-batching ($b = 1$) on the *ijcnn1* data set.

(a) $\lambda = 10^{-1}$

(b) $\lambda = 10^{-3}$

Figure 7: Comparison of theoretical variants of SVRG without mini-batching ($b = 1$) on the *real-sim* data set.

### G.1.2 Experiment 1.b: optimal mini-batching

Here we use the optimal mini-batch sizes we derived for *Free-SVRG* in Table 1 and *L-SVRG-D* in (57). Since the original SVRG theory has no analysis for mini-batching, and the current existing theory shows that its total complexity increases with $b$, we use $b = 1$ for SVRG. Like in Section G.1.1, the inner loop length is set to $m = n$. We confirm in these experiments that setting the mini-batch size to our predicted optimal value $b^*$ doesn't hurt our algorithms' performance. See Figures 8, 9, 10 and 11. Note that in Section G.2.2, we further confirm that $b^*$ outperforms multiple other choices of the mini-batch size. In most cases, *Free-SVRG* and *L-SVRG-D* outperform the vanilla SVRG algorithm both on the epoch and time plots, except for the regularized logistic regression on the *real-sim* data set (see Figure 11), which is a very easy problem since it is well conditioned. Comparing Figures 5 and 9 clearly underlines the speed improvement due to optimal mini-batching, both in epoch and time plots.

(a) $\lambda = 10^{-1}$

(b) $\lambda = 10^{-3}$

Figure 8: Comparison of theoretical variants of SVRG with optimal mini-batch size $b^*$ when theoretically available on the *YearPredictionMSD* data set.

(a) $\lambda = 10^{-1}$

(b) $\lambda = 10^{-3}$

Figure 9: Comparison of theoretical variants of SVRG with optimal mini-batch size $b^*$ when theoretically available on the *slice* data set.

(a) $\lambda = 10^{-1}$

(b) $\lambda = 10^{-3}$

Figure 10: Comparison of theoretical variants of SVRG with optimal mini-batch size $b^*$ when theoretically available on the *ijcnn1* data set.

(a) $\lambda = 10^{-1}$

(b) $\lambda = 10^{-3}$

Figure 11: Comparison of theoretical variants of SVRG with optimal mini-batch size $b^*$ when theoretically available on the *real-sim* data set.

### G.1.3 Experiment 1.c: theoretical inner loop size or update probability without mini-batching

Here, using $b = 1$, we set the inner loop size for *Free-SVRG* to its optimal value $m^* = 3L_{\max}/\mu$ that we derived in Proposition 6.1. We set $p = 1/m^*$ for *L-SVRG-D*. The inner loop length is set like in Section G.1.1. See Figures 12, 13, 14 and 15. By setting the size of the inner loop to its optimal value $m^*$, the results are similar to the one in experiments 1.a and 1.b. Yet, when comparing Figure 5 and Figure 13, we observe that it leads to a clear speed up of *Free-SVRG* and *L-SVRG-D*.

Figure 12: Comparison of theoretical variants of SVRG with optimal inner loop size $m^*$ when theoretically available ($b = 1$) on the *YearPredictionMSD* data set.

Figure 13: Comparison of theoretical variants of SVRG with optimal inner loop size $m^*$ when theoretically available ($b = 1$) on the *slice* data set.

Figure 14: Comparison of theoretical variants of SVRG with optimal inner loop size $m^*$ when theoretically available ($b = 1$) on the *ijcnn1* data set.

Figure 15: Comparison of theoretical variants of SVRG with optimal inner loop size $m^*$ when theoretically available ($b = 1$) on the *real-sim* data set.

## G.2    Optimality of our theoretical parameters

In this series of experiments, we only consider *Free-SVRG* for which we evaluate the efficiency of our theoretical optimal parameters, namely the mini-batch size $b^*$ and the inner loop length $m^*$.

### G.2.1    Experiment 2.a: comparing different choices for the mini-batch size

Here we consider *Free-SVRG* and compare its performance for different batch sizes: the optimal one $b^*$, 1, 100, $\sqrt{n}$ and $n$. In Figure 16, 17, 18 and 19, we show that the optimal mini-batch size we predict using Table 1 always leads to the fastest convergence in epoch plot (or at least near the fastest in Figure 16b).

(a) $\lambda = 10^{-1}$

(b) $\lambda = 10^{-3}$

Figure 16: Optimality of our mini-batch size $b^*$ given in Table 1 for *Free-SVRG* on the *YearPredictionMSD* data set.

(a) $\lambda = 10^{-1}$

(b) $\lambda = 10^{-3}$

Figure 17: Optimality of our mini-batch size $b^*$ given in Table 1 for *Free-SVRG* on the *slice* data set.

(a) $\lambda = 10^{-1}$

(b) $\lambda = 10^{-3}$

Figure 18: Optimality of our mini-batch size $b^*$ given in Table 1 for *Free-SVRG* on the *ijcnn1* data set.

(a) $\lambda = 10^{-1}$

(b) $\lambda = 10^{-3}$

Figure 19: Optimality of our mini-batch size $b^*$ given in Table 1 for *Free-SVRG* on the *real-sim* data set.

### G.2.2 Experiment 2.b: comparing different choices for the inner loop size

We set $b = 1$ and compare different values for the inner loop size: the optimal one $m^*$, $L_{\max}/\mu$, $3L_{\max}/\mu$ and $2n$ in order to validate our theory in Proposition 6.1, that is, that the overall performance of *Free-SVRG* is not sensitive to the range of values of $m$, so long as $m$ is close to $n$, $L_{\max}/\mu$ or anything in between. And indeed, this is what we confirmed in Figures 20, 21, 22 and 23. The choice $m = 2n$ is the one suggested by [13] in their practical SVRG (Option II). We notice that our optimal inner loop size $m^*$ underperforms compared to $n$ or $2n$ only in Figure 23a, which is a very rare kind of problem since it is very well conditioned ($L_{\max}/\mu \approx 4$).

Figure 20: Optimality of our inner loop size $m^* = 3L_{\max}/\mu$ for *Free-SVRG* on the
*YearPredictionMSD* data set.

Figure 21: Optimality of our inner loop size $m^* = 3L_{\max}/\mu$ for *Free-SVRG* on the *slice* data set.

(a) $\lambda = 10^{-1}$

(b) $\lambda = 10^{-3}$

Figure 22: Optimality of our inner loop size $m^* = 3L_{\max}/\mu$ for *Free-SVRG* on the *ijcnn1* data set.

(a) $\lambda = 10^{-1}$

(b) $\lambda = 10^{-3}$

Figure 23: Optimality of our inner loop size $m^* = 3L_{\max}/\mu$ for *Free-SVRG* on the *real-sim* data set.