[Reviews · NeurIPS 2019]

Reviewer 1



============================================================== After authors' response: the correction in Eq.(5) makes sense to me. I'm not very familiar with the mini-batch analysis in SVRG. It seems to me after reading the response that although the analysis for "arbitrary sampling" may not be that innovative, the analysis of mini-batch is new and different from previous works. From this perspective, I do agree the paper has its unique contribution. ============================================================== The paper is written in a clean way, and easy to read. From these perspective, I enjoyed reading this paper a lot. My major concerns are as stated in the contribution section: 1. Results and technics are not super innovative, similar analysis for other sampling scheme or non-averaging inner loop are done in [1, 10] and many other papers. Although they are on different settings, but the idea and proof shares a lot similarity. In this sense, I do not view the theoretical contribution of this paper as significant. 2. As this paper is targeted toward closing the gap of theory and practice, the slight weakness of theoretical results would not be a problem if this paper indeed performs practical experiment and showing the advantages of the changes. However, it's only on UCI dataset, on strongly convex loss functions. Some observations in experiments are not explained clearly. For instance, why standard SVRG (blue curve) would get stuck in a bad position for such a long time? It seems to be it's only because a very simple reason the choice of m is too large, so that standard SVRG use a very bad estimate of gradient for a very long time. However, this should not be an issue in practice, if practitioner tune the size of inner loop size, instead of blindly follow the conservative suggestion of theory. Finally, I feel the suggestion on the optimal minibatch size b (equation 5) a bit confusing... In fact, when m=n, and n<< L/mu, choosing b=1 gives complexity n + L_max/mu while choosing b=n gives complexity nL/mu. I'm wondering why the optimal choice of b is n in this case, especially when L_max = L? Do authors implicitly consider the setting L_max >> L?

Reviewer 2



######################################################## After reading the author's response, I retract my concern about knowledge of the strong convexity parameter. Regarding Figure 3, I am inclined to say that the wall clock plot doesn't seem to add anything to the discussion. Because b=1 is so slow, you can't see anything that is happening with the other lines, and in some sense this seems like an argument for why we should just ignore your paper and use any minibatch size we want as long as it's larger than 1. I would suggest either 1) removing the b=1 line from the plot so we can see what is happening with everything else or 2) removing the plot entirely. ######################################################## This is a nicely written paper, which extends existing theory of SVRG to cover parameter choices that are commonly made in practice. The proposed algorithms are not a dramatic improvement over SVRG, and they do not claim to be. However, they are somewhat simpler aesthetically, they decouple some of the hyperparameter choices to be more flexible, and the theory applies to any parameter choices. I see this paper mostly as reassurance for people who use SVRG that the simplifying choices they make (e.g. choosing m = n, sampling without replacement, etc) do not significantly hurt the theory behind the algorithm. I am not convinced that the ability to optimize b and or m according to the theory would be terribly useful in practice, but it does give some general guidance to what you should do e.g. when Lmax >> L versus when Lmax \approx L.

Reviewer 3



### post-rebuttal ### The authors did not indicate whether they will allude to related work mentioned. The papers mentioned indeed are similar (granted, the paper deals with finite sums whereas the references consider the streaming variant of the question). As a final note, the second reference does consider the precise question of what is the best batch size on every instance of a streaming least squares problem for mini-batch SGD - something that is similar to issues considered in this paper in the context of SVRG, aside from considering parallel/distributed SGD methods. ################# I like this paper's motivation, message and result. Aside from the comments below, I request the authors to go over typos/bugs (if any). The paper misses out on some references that have considered similar issues: [1] The work of Streaming SVRG - Frostig et al. indicates that the inner loop of SVRG needs to be run for a condition number of steps - where the notion of condition number is similar to one described in this paper. [2] The work of Jain et al (2018) "parallelizing stochastic gradient for least squares regression: mini-batching, averaging and model misspecification", JMLR. This paper studies mini-batching in the case of least squares regression and SGD. They have similar notions of expected smoothness, the best batch size etc., specialized to the case of least squares, which I think have been discussed in a very thorough manner.

[Author Response · NeurIPS 2019]

**Dear reviewers**, thank you for taking the time to review our paper and your careful remarks. We have addressed all of your questions, remarks and suggested improvements below.

**Rev. #2: Correction in Eq. (5).** We thank the reviewer for pointing out this inconsistency, it is in fact due to a typo in Eq. (5). When $n \leq L/\mu$ the optimal mini-batch size $b^*$ should be $b^* = \max\{\hat{b}, 1\}$ instead of $n$, where $\hat{b} := \sqrt{\frac{n}{2} \frac{L_{\max} - L}{nL - L_{\max}}}$. This fixes the inconsistency that the reviewer noted, indeed when $L = L_{\max}$ we have that $b^* = 1$.

**Smoothness and rate improvements.** In the case where $L = L_{\max}$, such as the extreme case of ridge regression with a diagonal matrix, there is indeed no benefit in mini-batching. But in general $L \leq L_{\max} \leq nL$ (see Lemma A.6), and in particular when $n$ is large $L_{\max}$ can be orders of magnitude greater than $L$. Indeed, all the benefits of mini-batching come from the gap between $L$ and $L_{\max}$, which is why any effective mini-batch analysis needs to leverage this gap and our analysis is the first to do so in this setting.

**Meaning of** *"practice of SVRG"* **in the title.** We propose and analyse algorithms which are as close as possible to what is already used in practice. In particular, we show that the commonly used parameter setting $m = n$ together with not resetting the inner iterates give the best rates one could expect in our setting. We do not claim to establish a new *defacto* practice, but rather to better understand the SVRG that is already implemented and used for generalized linear models[1]

**Innovative nature of our results and analysis.** Our analysis cracks the open question of showing the benefits for mini-batching for SVRG. We do this by using a new Lyapunov-style analysis. It is this novelty that allows us to freely set the inner loop length $m$, but also understand mini-batching. Moreover, neither the theory for the non-convex setting [10] nor the accelerated SVRG [1] are able to show the benefits of mini-batching. In particular, it is shown in [1] that the iteration complexity benefits from a larger mini-batch size, but not the total complexity. And [10] determines the optimal importance sampling strategy depending on a batch size fixed *a priori*.

**Behaviour of SVRG in Figure 2.** The purpose of this experiment was to see which of the settings guaranteed to work in theory for SVRG resulted in the best practical performance. That is, we compared the settings of the previous theory (see Theorem 6.5 of Bubeck, Convex Optimization Algorithms and Complexity, 2015) that suggests to set $m = 20L_{\max}/\mu$ (standard SVRG is the blue curve) and our theory which suggests that any choice for $m$ between $3L_{\max}/\mu$ and $n$ will suffice. Because $m$ is so large for the standard SVRG, we can see the method stalling. Thus the experiment carefully highlights the issues with the current standing theory of SVRG. Though one could always resort to using a grid search to determine parameters such as $m$, the stepsize or the mini-batch size, the purpose of our work is to exactly avoid the elevated costs related to performing such grid searches.

**Usefulness in large-scale setting.** The setting of our paper is for strongly convex learning problems. Both CIFAR and ImageNet are not appropriate benchmarks in our setting since convex models are not able to properly fit or make predictions over these data sets, independently of the optimization method used. The main applications of SVRG are medium-scale convex learning problems, such as those that we solved: regularized logistic regression on *ijcnn1* ($n = 141, 691, d = 22$) and *real-sim* ($n = 72, 309, d = 20, 958$), and ridge regression on *YearPredictionMSD* ($n = 515, 345, d = 90$) and *slice* ($n = 53, 500, d = 384$). We will make sure to clarify this in our final submission.

**Rev. #4: Benefit of providing optimal $b$ and $m$.** Since our submission, we have performed extensive numerical experiments comparing our optimal mini-batch size $b^*$ against the empirical best mini-batch size over a large grid. We find that the resulting empirical complexity of using our $b^*$ is remarkably close to the best one over the grid search. We will include these new experiments in our final submission. In particular, we are now able to predict the optimal mini-batch size even more accurately due to some minor improvements we have made in our theory (we have shaved off some constants). We will also include this update on the improved theory in our final submission.

**Free-SVRG needs $\mu$, not L-SVRG-D.** In Algorithm 1, the reference point is reset to an exponentially weighted average of the inner iterates. The weights $p_t$ depend on the strong convexity constant (see Eq. (11) and Algo. 1). This motivated us to present and analyse a single loop version of SVRG, which does not require $\mu$ at any point. We will make sure to clarify this point in our discussion.

**Comment on Fig. 3.** The experiments were run on a multi-core server. The difference between some of the epoch and time plots is due to the mini-batch methods benefiting from shared memory parallelism. Unfortunately, this is always an issue with plotting time, as it depends on the architecture used. We will include information about the architecture in the supplementary material and explain this apparent discrepancy.

**Our definition of complexity.** We will clarify that the symbol $C_m(b)$ is in fact an upper bound on the total complexity.

**Rev. #5:** We thank you for pointing out these references. Streaming SVRG (Frostig et al.) presents some interesting ideas: they provide statistical guarantees for a version of SVRG in the infinite samples framework, and a schedule of increasing mini-batch sizes. Yet their assumptions and objectives are different from ours. As for Jain et al. (2018), their analysis uses a similar notion of expected smoothness, but they only analyse quadratics and parallelised SGD, which is a completely different setting.

## Footnotes

[1]See for example the lightning package from scikit-learn [20]: http://contrib.scikit-learn.org/lightning/


[Meta-Review · NeurIPS 2019]

All the reviewers agreed that this is a nice paper that gives some theoretical grounding for some of the empirical choices made when applying SVRG in practice. In particular, they analyze smaller epochs (m=n) and non-iid sampling/batching (e.g. minibatch without replacement), and they analyze an algorithm where the first iterate of an epoch is the last iterate of the previous one rather than an average over the previous epoch. While individually, these contributions are somewhat modest, together they paint a fairly complete picture of SVRG that will be useful for the community.